# Half-metre sea-level fluctuations on centennial timescales from mid-Holocene corals of Southeast Asia

Aron J. Meltzner[1], Adam D. Switzer[1,2,3], Benjamin P. Horton[1,2,4,5], Erica Ashe[5,6], Qiang Qiu[1,2], David F. Hill[7], Sarah L. Bradley[8,9], Robert E. Kopp[5,10,11], Emma M. Hill[1,2], Jędrzej M. Majewski[1,2], Danny H. Natawidjaja[12] & Bambang W. Suwargadi[12]

Sea-level rise is a global problem, yet to forecast future changes, we must understand how and why relative sea level (RSL) varied in the past, on local to global scales. In East and Southeast Asia, details of Holocene RSL are poorly understood. Here we present two independent high-resolution RSL proxy records from Belitung Island on the Sunda Shelf. These records capture spatial variations in glacial isostatic adjustment and paleotidal range, yet both reveal a RSL history between 6850 and 6500 cal years BP that includes two 0.6 m fluctuations, with rates of RSL change reaching $13 \pm 4$ mm per year ($2\sigma$). Observations along the south coast of China, although of a lower resolution, reveal fluctuations similar in amplitude and timing to those on the Sunda Shelf. The consistency of the Southeast Asian records, from sites 2,600 km apart, suggests that the records reflect regional changes in RSL that are unprecedented in modern times.

[1] Earth Observatory of Singapore, Nanyang Technological University, 50 Nanyang Avenue, Singapore 639798, Singapore. [2] Asian School of the Environment, Nanyang Technological University, 50 Nanyang Avenue, Singapore 639798, Singapore. [3] Complexity Institute, Nanyang Technological University, 50 Nanyang Avenue, Singapore 639798, Singapore. [4] Department of Marine and Coastal Sciences, Rutgers University, 71 Dudley Road, New Brunswick, New Jersey 08901, USA. [5] Institute of Earth, Ocean and Atmospheric Sciences, Rutgers University, New Brunswick, New Jersey 08901, USA. [6] Department of Statistics and Biostatistics, Rutgers University, 110 Frelinghuysen Road, Piscataway, New Jersey 08854, USA. [7] School of Civil and Construction Engineering, Oregon State University, 1491 SW Campus Way, Corvallis, Oregon 97331, USA. [8] Institute for Marine and Atmospheric Research, Utrecht University, Princetonplein 5, 3584 CC Utrecht, Netherlands. [9] Department of Geoscience and Remote Sensing, Delft University of Technology, Stevinweg 1, 2628 CN Delft, Netherlands. [10] Department of Earth and Planetary Sciences, Rutgers University, 610 Taylor Road, Piscataway, New Jersey 08854, USA. [11] Rutgers Energy Institute, Rutgers University, New Brunswick, New Jersey 08901, USA. [12] Research Center for Geotechnology, Indonesian Institute of Sciences (LIPI), Kompleks LIPI, Gedung 70, Jalan Sangkuriang, Bandung, Jawa Barat 40135, Indonesia. Correspondence and requests for materials should be addressed to A.J.M. (email: meltzner@ntu.edu.sg).

More than 100 million people, mostly in East and Southeast Asia, live within 1 m of sea level and are acutely susceptible to sea-level rise brought about by climate change[1]. Regional sea-level change is a superposition of secular eustatic trends and interannual regional oscillations, not all of which are well studied. The largest interannual variability of sea level occurs in the tropical Pacific and is related to the El Niño–Southern Oscillation (ENSO); early (1993–2001) satellite data showed high rates of sea-level rise in Southeast Asia that approached 30 mm per year[2], though those extreme rates have not persisted[3,4].

Understanding the extent to which sea-level changes in East and Southeast Asia are affected by interannual sea-level variations is important to protecting vulnerable coastal assets in low-lying deltas[5] and atoll islands[6]. But how interannual sea-level fluctuations will change in association with a projected increase in extreme ENSO and other patterns of atmosphere/ocean variability due to climate change remains unknown[7]. Proxy-based paleo-sea level reconstructions characterize patterns of natural variability and provide a target for calibrating models of the relationship between climate and sea level, as well as a pre-Industrial background against which to compare recent trends[8]. These proxy reconstructions, however, have hitherto been hindered by accuracy and precision, particularly in East and Southeast Asia[9].

One relative sea level (RSL) proxy that has seen limited use in East and Southeast Asia is coral microatolls. Microatolls track RSL with accuracy and high precision. Prolonged subaerial exposure at times of extreme low water restricts the highest level to which the coral colonies can grow[10–15]. Portions of the coral living above this elevation die during a period of extreme low water, but portions below this continue to grow outward (and upward) until the next incidence of extreme low water. A microatoll's concentric annuli form as a result of this repeating sequence of slow upward growth and sudden diedowns, superimposed on longer-term RSL trends[15].

With regard to microatolls, the term diedown refers to a partial mortality event on a coral colony in which the portion of a coral above a certain elevation dies, while coral polyps at lower elevations survive. Unlike a complete mortality of a coral colony, for which the interpretation of the cause of death is not always straightforward, a diedown to a uniform elevation around the perimeter of the coral is a clear indication that the diedown resulted from low water. The elevation above which all coral died is termed the highest level of survival (HLS)[11]. A related term, the highest level of growth (HLG)[14], reflects the highest elevation up to which a coral grew in a given year. Although both HLS and HLG refer to the highest living coral at a particular time of interest, HLG is limited by a coral's upward growth rate. Hence, in years during which no diedown occurs, HLG provides only a minimum estimate of the HLS that would theoretically be possible.

A microatoll's basic morphology reveals important information about RSL during the coral's lifetime. Although a fall in RSL that triggers a diedown might be very short lived, such as during a single extreme low tide, multi-decadal trends in RSL can be established by comparing the elevations of several successive diedowns. Flat-topped microatolls record RSL stability; colonies with diedowns (HLS unconformities) that rise radially outward towards their perimeter reflect rising RSL during their decades of growth; corals with progressively lower diedowns reflect falling RSL. As RSL rises and falls over time, microatoll morphologies record these changes in RSL. Because these corals' skeletons have annual growth bands—a result of the contrasting density in growth at different times of the year—we can precisely count the years over which these changes occur.

We derive proxy records of mid-Holocene RSL from coral microatolls at two sites on Belitung Island, Indonesia, on the Sunda Shelf: TBAT, in the southeast; and TKUB, 80 km to the northwest (Fig. 1; Supplementary Table 1). To extract climate-related rates of RSL change, we chose a region that is inferred to be tectonically stable[16], and sites where abundant granitic outcrops suggest minimal sediment compaction. The Indonesian proxy records reveal 0.6-m swings in RSL over several centuries during the mid-Holocene. Accounting for systematic shifts in elevation between the time series at the two sites, and for peculiarities of microatoll growth over the 18.61-year nodal tidal cycle, we use a hierarchical statistical model to show that a substantial majority of the multi-decadal scale fluctuations observed in each data set can be explained by a shared sea-level curve. Consideration of reinterpreted data from an earlier study[17], which suggest coeval fluctuations of a similar amplitude 2,600 km away along the southern coast of China, argues that these changes were at least regional in scope.

## Results

**Microatoll growth over the nodal tidal cycle on the Sunda Shelf.** The tidal range is modulated over the 18.61-year nodal tidal cycle. When lunar declination is at a maximum, such as in 2006 and 2025, the range of the predominantly diurnal tides near Belitung is 13–14% greater than average; when lunar declination

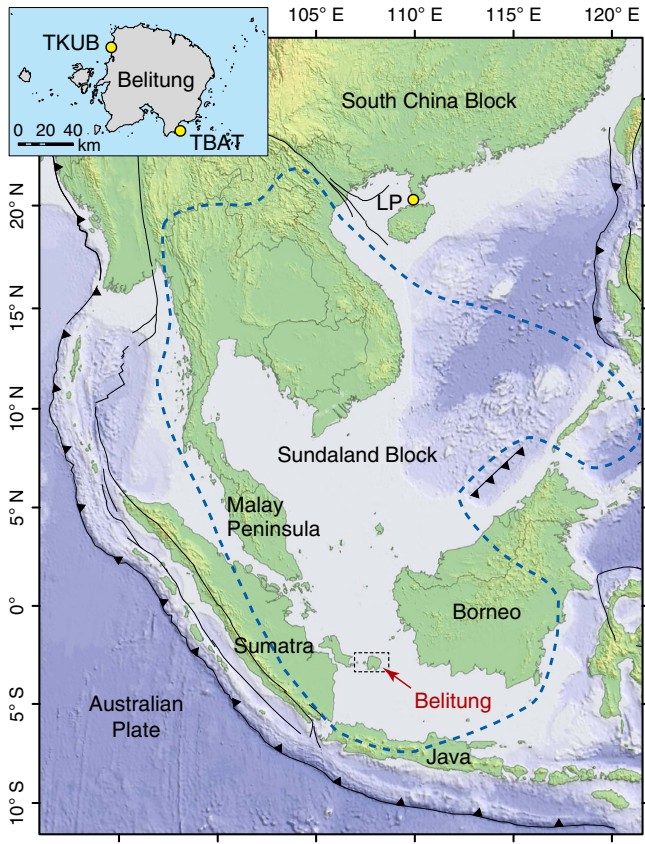

**Figure 1 | Map of the stable Sundaland block showing site locations.** The extent of Sundaland, encompassed by the dashed blue curve, is approximated as the region deforming horizontally at <4 mm per year relative to its core[16]. Yellow circles mark coral microatoll study sites: LP, Leizhou Peninsula site[17]; TKUB and TBAT (shown in inset), Tanjung Kubu and Tanjung Batuitam sites presented here. The inset location is denoted on the main map by a dashed box. Solid lines depict primary faults; barbed lines represent traces of subduction zones. Map created in ArcGIS.

is at a minimum, such as in 2015, the tidal range near Belitung is 13–14% smaller than average. Given the 2.9-m overall tidal range along northwestern Belitung, the net effect of this is that the lowest tide at the TKUB site in 2006 was 0.35 m lower than the lowest tide in 2015 (Fig. 2a).

Because microatoll HLS is governed by extreme low water, diedowns tend to occur during portions of the 18.61-year cycle when the year-to-year lowest tides are becoming increasingly low, or when the tides are near their lowest levels. Subsequently, the year-to-year lowest tides rise more rapidly than corals can grow up. For instance, if a microatoll at TKUB experienced a diedown during the lowest tide in 2006 and grew upward at ∼15 mm per year thereafter, it would have grown 0.15 m vertically over the following 10 years, but its highest coral polyps (its HLG) would have been 0.20 m lower than the theoretical HLS in 2016 (Fig. 2c). It would only be in 2020 that the upward coral growth would catch up to HLS, and with the lowest tides falling each year from 2020 until 2025, diedowns would be expected only in that interval. Similarly, diedowns at TKUB would have been expected in roughly the intervals 2001–2006, 1982–1987, 1964–1969 and so forth in the past. Transient meteorological conditions (such as rain, wind or cloud cover) and longer-term hydroclimatic oscillations (such as ENSO) also influence local sea level and coral diedowns, so minor deviations are expected in the actual timing and amplitude of the diedowns (for example, Fig. 2d–g).

As proxies for sea level, we consider diedowns (HLS elevations) as sea-level index points. A sea-level index point estimates the unique position of RSL in space and time[18]. HLG data are minimum limiting data, as they provide only a minimum bound on the theoretical HLS in a particular year (see Methods). A perhaps counterintuitive consequence of microatoll growth over the 18.61-year tidal cycle is that the highest minimum limiting points in each cycle are expected to be up to 0.2 m higher than the lowest index points in the same cycle, even if RSL is stable over that interval. Any modelling methodology must account for this expected periodicity, and interannual trends and rates of RSL change may be meaningful only when averaged over periods longer than one 18.61-year cycle.

**Chronological constraints and uncertainties**. We distinguish three kinds of chronological uncertainty in our study, and we treat the coral records as floating chronologies with appropriate constraints from radiocarbon dating. First, the relative age uncertainty between two parts of an individual coral slab is simply the annual band-counting uncertainty, which is commonly less than ±1 year. At the TBAT site, where the entire record comes from a single coral, relative age uncertainties are all in this category. In cases where two slabs have overlapping calibrated radiocarbon age estimates and matching diedown chronologies, those slabs can be coupled together as a single floating chronology, and the relative age uncertainty between various parts of those slabs is also determined from the band-counting uncertainty[14,19–21]; this is the case for some of the TKUB corals, as discussed later.

Second, the relative age uncertainty between distinct corals at an individual site is governed by calibrated radiocarbon age errors. For radiocarbon dating of marine samples such as corals, a marine calibration curve is used[22], and every site has a localized marine reservoir correction, $\Delta R$, expressed as an offset (in years) from a global-mean value. Although site-specific $\Delta R$ values typically have uncertainties of decades to centuries, we extracted multiple samples from each slab for dating, and the exceptional consistency between the redundant dates on each coral (Supplementary Table 2) indicates that the $\Delta R$ value at each site did not vary temporally over the period of study (see Methods). We can,

therefore, ignore uncertainties in $\Delta R$ if we are interested only in the relative age of two corals at the same site. At the TKUB site, relative age uncertainties between the corals do not exceed 70–80 years ($2\sigma$; Supplementary Table 2). To estimate absolute ages, we assumed $\Delta R \approx +89$ year, based on a nearby sample from southwestern Borneo[23], but our primary conclusions do not depend upon knowing this correction accurately.

Third, absolute ages for each RSL proxy time series carry additional uncertainty resulting from the unmodelled error in $\Delta R$. This uncertainty applies uniformly to each site's RSL time series as a whole, based on the argument that $\Delta R$ at each site remained constant over the period of study, affecting the absolute timing of each curve, but not its shape. The uncertainty may be ±85 year, based on the $\Delta R$ error of modern samples collected nearby[23] and mid-Holocene samples from the South China Sea[24]. While this absolute age uncertainty is not reflected in the dates reported in Supplementary Table 2 to facilitate comparison between different parts of each RSL record, readers should note that each site's RSL curve could be shifted uniformly by up to ±85 year.

**Vertical uncertainties of microatoll data**. To estimate vertical uncertainties, we surveyed living coral microatolls at both the TBAT (southeastern) and TKUB (northwestern) sites (see Methods). Ponding of water at low tide, particularly on a wide coral reef, is a known complication that allows individual corals to grow above the theoretical HLS[15]. We, therefore, considered a mix of ponded and open-ocean microatolls in our survey, classifying each colony as either clearly open-ocean, clearly ponded or possibly ponded. The result, shown in Fig. 3, represents the distribution of HLS elevations immediately following a diedown. HLG elevations in subsequent years would be higher than the elevations shown, by an amount dependent upon the coral growth rate and the time since the most recent diedown. The s.d. of modern HLS at each Belitung site, including ponded and open-ocean microatolls, is 0.09 m; we apply this as the error to the fossil (mid-Holocene) coral data as well.

**Microatoll slabs from the Belitung sites**. The RSL history of southeastern Belitung was reconstructed from a single, particularly long-lived coral microatoll, TBAT-F01. This microatoll, and others nearby that were inferred to belong to the same generation, had a high central dome surrounded by low middle concentric annuli and high outer annuli (Fig. 4). This structure requires growth under oscillating RSL conditions. We extracted two radial slabs from microatoll TBAT-F01 (Fig. 5), and we used both slabs to constrain the site's RSL history (Fig. 6). The annual banding visible in X-rays of the slabs indicated the coral grew for >240 years. Radiocarbon ages, if taken at face value, indicate the coral tracked RSL from 6,750 to 6,530 cal years BP, though erosion of the outermost part of the coral has rendered the youngest ∼40 years of growth less useful. The coral's central dome grew upward to 1.0 m higher than analogous corals living today, suggesting that the amplitude of the first RSL peak was at least +1.0 m (above modern levels); however, the coral had not yet grown up to its HLS before RSL rapidly fell, so the actual height of the first peak is unknown and may have been higher. The first RSL peak had occurred by 6,750 cal years BP. RSL then fell to +0.6 m, remaining at a lowstand for ∼80 years, before rising to a second peak at +1.2 m shortly after 6,600 cal years BP.

The RSL history of northwestern Belitung was recorded collectively by five shorter-lived corals at different elevations. Slabs from these microatolls (TKUB-F04, TKUB-F05, TKUB-F16, TKUB-F19 and TKUB-F23) appear in Fig. 7. Analyses of slab growth patterns and radiocarbon dates from each microatoll suggest that TKUB-F04 and TKUB-F05 were coeval and

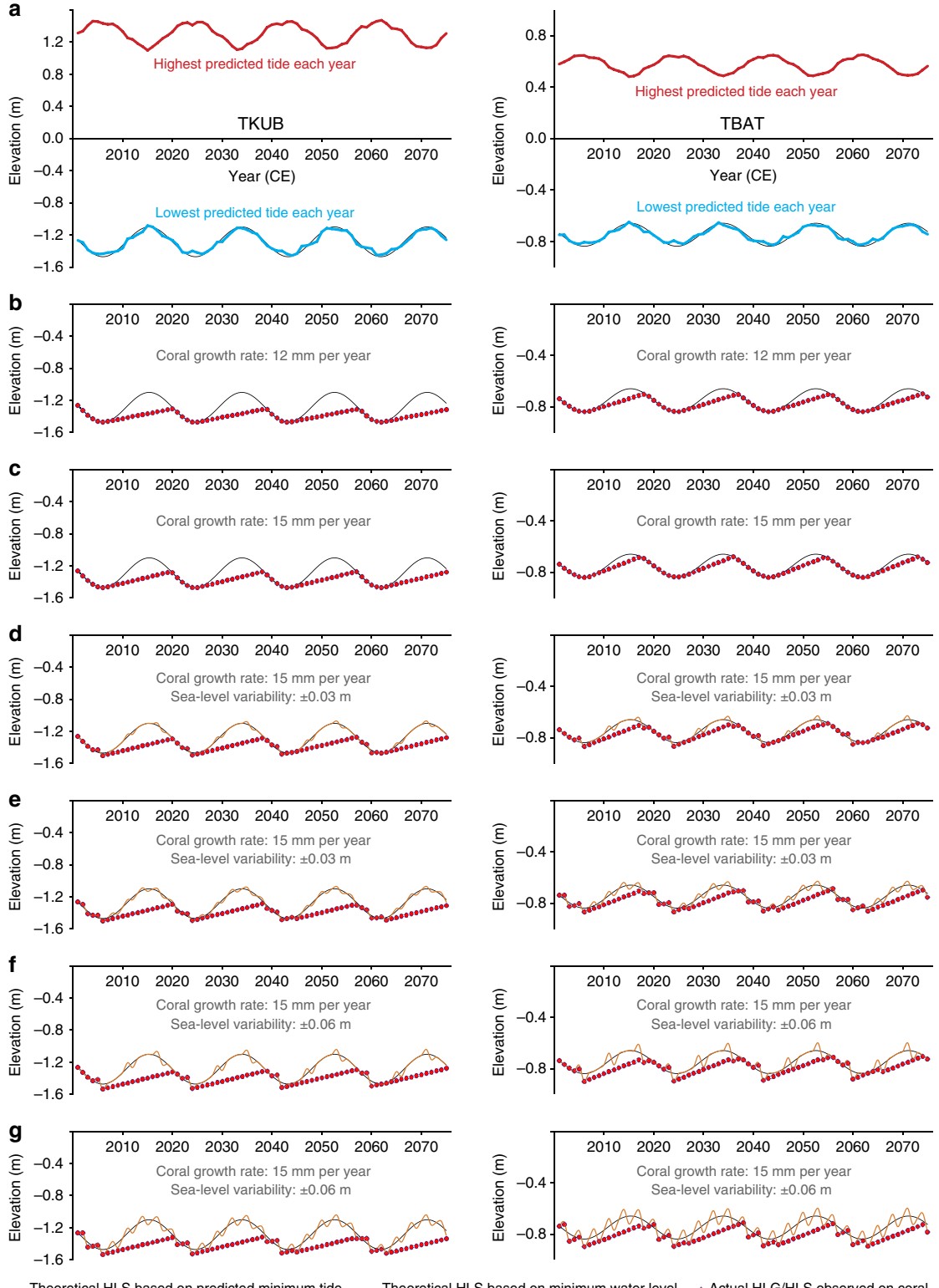

**Figure 2 | Schematic coral growth scenarios at the TKUB and TBAT sites.** The observed propensity for and occasional clustering of coral diedowns toward the end of each 18.61-year nodal tidal cycle is predicted by schematic models of coral growth over the 18.61-year cycle. These models illustrate the year-to-year variability of the difference between the highest living coral polyps and the lowest water levels. For each panel, the situation at TKUB is on the left and the situation at TBAT is on the right. (**a**) Highest and lowest tides in each calendar year predicted for each site by a tidal model[39]. In black, we fit a sinusoid (fixed period: 18.61 years) to the annual lowest tides; that sinusoid is reproduced in **b** through **g** as the 'theoretical HLS based on predicted minimum tide'. (**b**,**c**) Expected highest level of growth (HLG) or highest level of survival (HLS) based on coral growth rates of 12 mm per year (**b**) or 15 mm per year (**c**) in light of the predicted annual minimum tides at each site. There is a nonzero vertical offset between coral HLS and minimum water level, but that offset is assumed to be constant over time and is ignored here for the sake of simplicity. (**d**,**e**) The scenario in **c**, but with the added complexity of interannual sea-level variability of ± 0.03 m. (**f**,**g**) The scenario in **c**, but with the added complexity of interannual sea-level variability of ± 0.06 m.

constitute a single floating chronology, while TKUB-F16 and TKUB-F19 also overlapped in time and form a second floating chronology; the TKUB-F23 record, by itself, is a third floating chronology at the TKUB site. Our initial reconstruction of the RSL history of the TKUB site (Fig. 8) reflects radiocarbon ages calculated assuming $\Delta R = +89$ years, identical to the correction at the southeastern site (see Methods). The age of each coral is adjusted in this reconstruction by as much as a few decades to avoid inconsistencies among the five corals, but all coral ages as plotted in Fig. 8 remain within the $2\sigma$ limits of the modelled radiocarbon age errors in Supplementary Table 2. Again taken at face value, the resulting ages suggest the five corals grew between 6,800 and 6,440 cal years BP, with RSL fluctuations resembling those at southeastern Belitung. Here RSL rose to an initial peak of $+1.9$ at 6,720 cal years BP, then fell rapidly to a lowstand of $+1.3$ m, remaining at about that level for $\sim 100$ years, before rising to a second peak at $+1.7$ m shortly after 6,550 cal years BP. Around 6,480 cal years BP, RSL appears to have fallen again to $+1.3$ m before rising to a third peak at $+1.6$ m or higher. Within the uncertainties of the various $2\sigma$ radiocarbon age errors, all of the TKUB corals (or perhaps only the oldest floating chronology, TKUB-F04 and TKUB-F05) might be as much as 21 years older. In addition, if the marine radiocarbon reservoir correction, $\Delta R$, differed from the assumed value of $+89$ years, then there might be a uniform shift in all the dates from the site, within the uncertainty of $\Delta R$.

**Comparison and modelling of RSL proxy records.** A comparison of the time series from the two Belitung sites (Figs 6 and 8) reveals RSL fluctuations of similar amplitude at both sites, but with small, systematic shifts along both axes. First, the mid-Holocene HLS and HLG elevations at the northwestern site (TKUB) are consistently 0.5–0.7 m higher than 80 km to the southeast (at TBAT). Second, the fluctuations occur roughly half a century later at TKUB than at TBAT, if the radiocarbon ages are taken at face value and if $\Delta R = +89$ years at both sites.

Prompted by these striking similarities yet systematic differences, we sought to model the RSL proxy reconstructions as a combination of a shared non-linear signal and a site-specific offset, plus a periodic term to model microatoll growth over the 18.61-year tidal cycle. We constructed a hierarchical statistical model after Kopp *et al.*[25], separated into three levels: a data level, which models the recording of RSL by proxies; a process level, which models RSL at the different sites; and a hyperparameter level, which characterizes key attributes of the underlying levels. This model optimizes the relative timing of each floating chronology, subject to appropriate radiocarbon dating constraints. It also allows separation of the non-linear and periodic signals and the site-specific offset. Details are given in Methods. The optimized model appears in Fig. 9 with the chronologically optimized time series from the TKUB site.

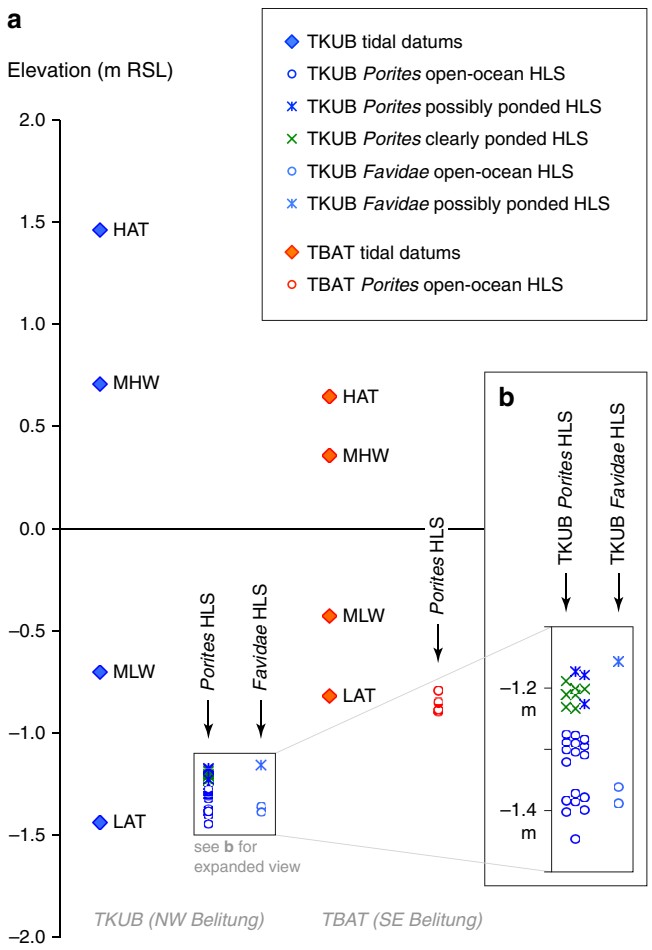

**Figure 3 | Living coral HLS at each site compared with tidal datums.** (**a**) Tidal datums, estimated from the Oregon State University regional tidal inversion for the Indian Ocean region[39], are shown relative to surveyed microatoll elevations at each site. HAT, highest astronomical tide; LAT, lowest astronomical tide; MHW, mean high water; MLW, mean low water. (**b**) Expanded view of the microatoll elevations at the TKUB site, displaying individual microatoll elevations. See Methods for details of the construction of this plot.

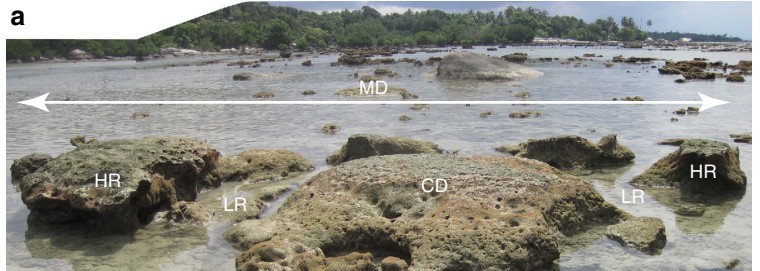

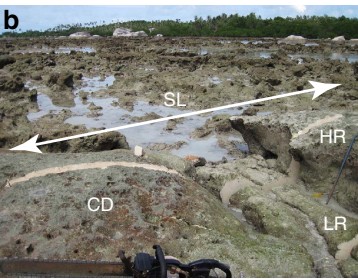

**Figure 4 | TBAT-F01 fossil microatoll at site TBAT.** The central dome grew during an initial RSL peak; RSL then dropped, forming several lower concentric rings; RSL then rose again, allowing the outer flank to grow upward. Two radial slabs were extracted from this microatoll and are shown in Fig. 5. (**a**) Photomosaic looking west-northwest, before slabbing. (**b**) Photo looking west-southwest, after TBAT-F01A was extracted and while TBAT-F01B was being cut. CD, central dome; HR, high outer concentric rings; LR, low middle concentric rings; MD, microatoll diameter (8 m, see arrows); SL, slab TBAT-F01A length (4.4 m, see arrows).

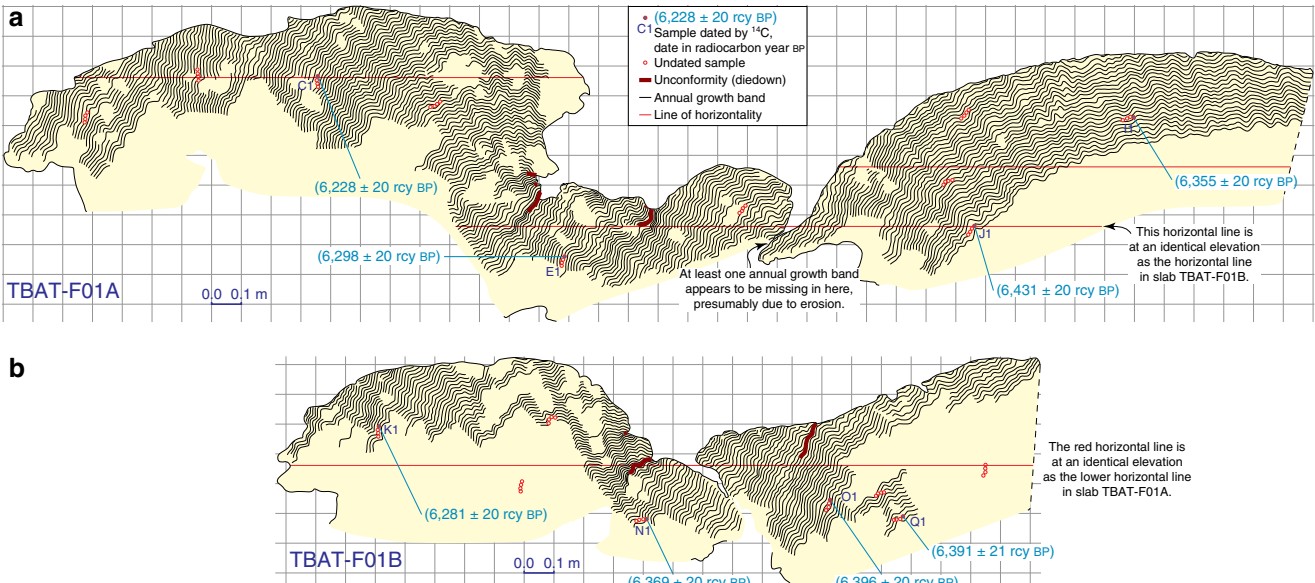

**Figure 5 | TBAT-F01 fossil microatoll slabs from site TBAT.** Annotated cross-sectional slabs (**a**) TBAT-F01A and (**b**) TBAT-F01B through the TBAT-F01 fossil microatoll at the southeastern Belitung site. In both, the central dome, which grew during the first peak, is to the right, although slab TBAT-F01B did not extend fully to the center of the microatoll. After the central dome grew, sea level dropped, forming a sequence of lower concentric rings in the middle of each cross-section. Sea level then rose again, allowing the outer flank to grow upward to the left in each slab. Annual banding is traced where it was clear in the X-rays. Conventional (uncalibrated) radiocarbon ages are reported with $1\sigma$ errors; see Supplementary Table 2 for calibrated ages.

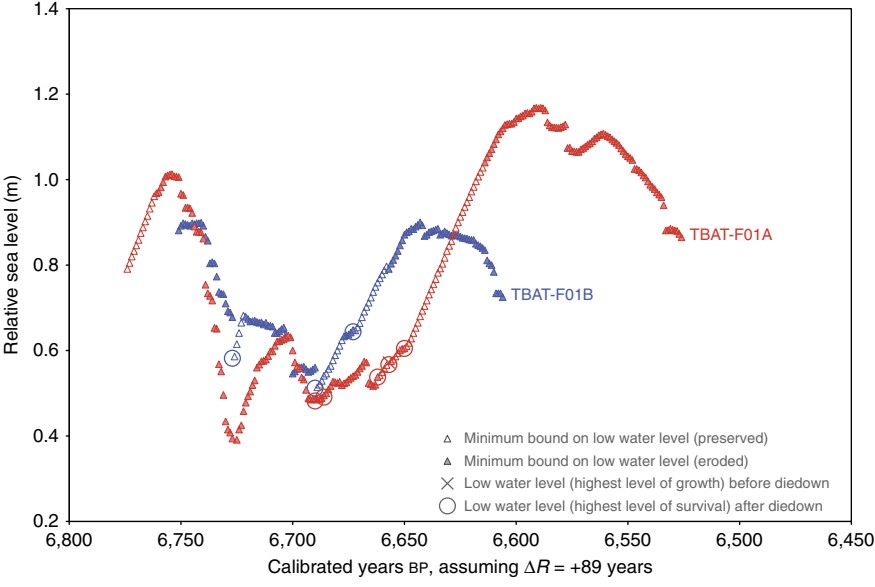

**Figure 6 | Mid-Holocene RSL proxy reconstruction from the TBAT-F01 microatoll at site TBAT.** Data from the two slabs are shown in different colours. Vertical uncertainties are uniformly $\pm 0.09$ m ($1\sigma$) but are not shown for clarity. The relative age uncertainty between two observations at this site is simply the annual band-counting uncertainty, commonly less than $\pm 1$ year; however, unmodelled uncertainty in $\Delta R$ could affect absolute ages and would allow the entire curve to be shifted uniformly by up to $\pm 85$ years. In particular, $\Delta R$ at TKUB may differ from $\Delta R$ at TBAT, contrary to our assumption. The timing of the data from this site was held fixed in our model; the relative timing of the floating chronologies at the TKUB site (Fig. 8) was ultimately optimized by the model.

**Shared RSL curve for Belitung Island**. We hold the timing of the TBAT curve fixed in the model optimization to that determined assuming $\Delta R = +89$ years. The TKUB curve is shifted 54 years older overall; the oldest floating chronology at TKUB (TKUB-F04 and TKUB-F05) is shifted an additional 20 years older relative to the central floating chronology (TKUB-F16 and TKUB-F19); and TKUB-23 is shifted 1 year younger relative to TKUB-F16 and TKUB-F19 (Supplementary Table 3). The 54-year shift of the overall TKUB curve is reasonable given that it is well within the

$\pm 85$-year unmodelled error in $\Delta R$, and the 20- and 1-year shifts between the floating chronologies at TKUB are at the limit of what is permitted by the uncertainties of the various $2\sigma$ calibrated radiocarbon age errors. Although the model has optimized the shifts between the floating chronologies at TKUB to total 21 years, we note that, even with somewhat smaller shifts, the model would still fit the data well.

Collectively, the corals provided 25 sea-level index points (HLS elevations following diedowns) and annual minimum limiting

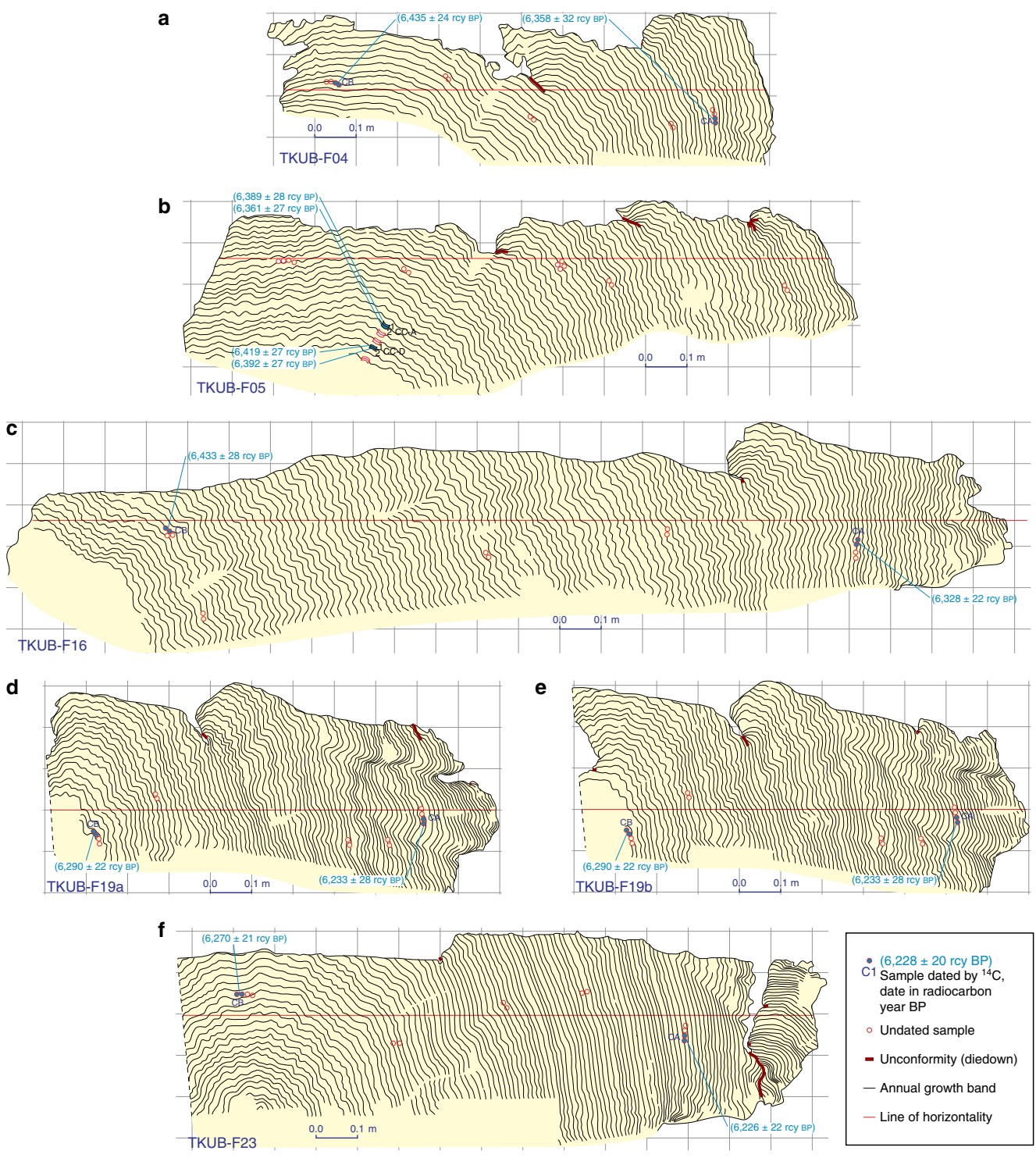

**Figure 7 | Fossil microatoll slabs from site TKUB.** Annotated cross-sections through fossil microatolls at the northwestern Belitung site: (**a**) slab TKUB-F04; (**b**) slab TKUB-F05; (**c**) slab TKUB-F16; (**d,e**) two parallel slices through slab TKUB-F19; (**f**) slab TKUB-F23. Annual banding is traced where it was clear in the X-rays. Conventional (uncalibrated) radiocarbon ages are reported with 1σ errors; see Supplementary Table 2 for calibrated ages.

data (minimum bounds on the theoretical HLS) for a span of >350 years (Fig. 9). The results suggest an initial RSL peak at ~6,800 cal years BP; RSL then fell ~0.6 m and remained at a lowstand for 80–100 years, before rising 0.4–0.6 m to a second peak at ~6,590 cal years BP. Corals at TKUB record a second drop at ~6,530 cal years BP, with a third peak shortly thereafter.

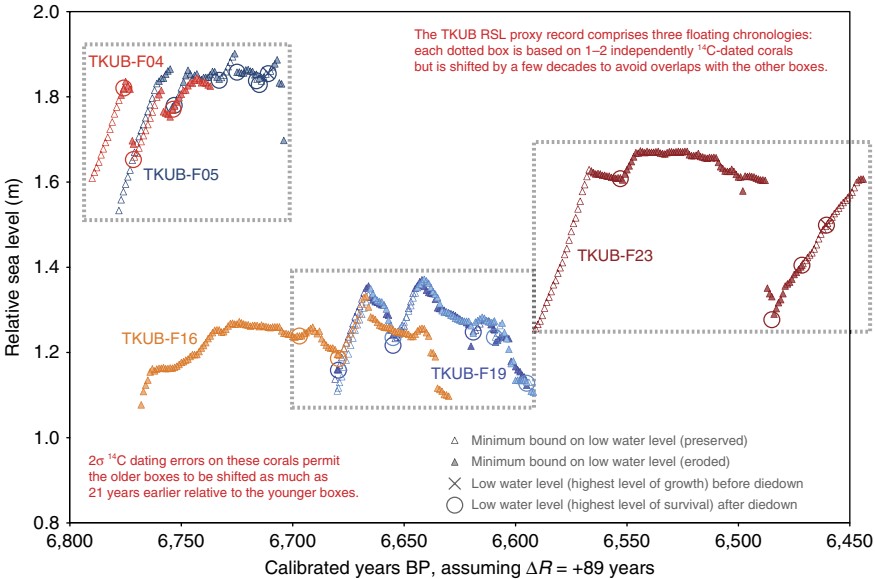

**Figure 8 | Mid-Holocene RSL proxy reconstruction from coral microtolls at site TKUB.** Colours correspond to data from different slabs. Vertical uncertainties are uniformly ± 0.09 m (1σ) but are not shown for clarity. This history consists of three discrete floating chronologies (FCs) that have been independently radiocarbon dated (each FC indicated by a dotted box). Data within each box are fixed to one another by the counting of annual growth bands, and have less than ± 1 year of relative age uncertainty; however, the radiocarbon dates permit as many as 21 years of additional separation between the oldest and youngest FCs. ΔR did not vary over the lifetime of these corals, and therefore uncertainty in ΔR can be ignored for determining the relative age of two corals at this site; however, unmodelled uncertainty in ΔR could affect absolute ages and would allow the entire curve to be shifted uniformly by up to ± 85 years. In particular, ΔR at TKUB may differ from ΔR at TBAT, contrary to our assumption. The relative timing of each FC was ultimately optimized by the model, as shown in Fig. 9.

Although no data exist from this later period at TBAT, this second drop in RSL could explain the death of TBAT-F01 at ~6,530 cal years BP.

The peak rate of RSL rise, averaged over a 20-year running time window over the period of study (~6,850–6,500 cal years BP), is + 9.6 ± 4.2 mm per year (2σ); the peak rate of RSL fall is − 12.6 ± 4.2 mm per year (Supplementary Table 4). If the 21-year shift between the floating chronologies at TKUB was reduced as contemplated in the previous paragraph, the peak rate of RSL fall (~6,770 cal years BP) would be even faster.

**Site-specific offset between TKUB and TBAT.** Several possible mechanisms could explain the systematically higher elevations at the TKUB site. The primary cause of this offset is the interplay between two processes of glacial isostatic adjustment (GIA) that drove RSL change at far-field sites during the mid-Holocene: equatorial ocean syphoning and continental levering[26,27]. Equatorial ocean syphoning results in far-field RSL fall, due to the migration of water from the far field to the near field to fill the space vacated by the collapsing forebulge at the periphery of previously glaciated regions. Continental levering from increased ocean load along continental margins induces uplift at inland regions and subsidence within the ocean basin, generating large sea-level gradients perpendicular to the coast (as shown on Supplementary Fig. 1). The spatially complex signal resulting from the two larger nearby landmasses of Sumatra (to the west) and Borneo (to the east) drive a pronounced difference in the RSL signal across Belitung. This inference is supported by a recently developed GIA model for Southeast Asia, which shows that at ~6.5 kyr BP, RSL should be higher on northwestern Belitung than at the southeastern site by ~0.2–0.4 m, regardless of the choice of earth model and ice model (Methods; Supplementary Fig. 1; Supplementary Table 5)[28]. In addition to the effects of GIA, a small portion (~0.1 m) of the systematic difference in

elevations between the sites could result from changes in the tidal range[29]. The tidal range at both sites is modelled to have been slightly higher at 6–7 kyr BP, implying that mean sea level was up to ~0.1 m higher than shown on Fig. 9; this effect would have been more pronounced at TBAT than at TKUB (Methods; Supplementary Fig. 2).

A number of processes that have been observed at microatoll study sites elsewhere can be excluded at our sites. Significant compaction (> 0.1 m) is not a factor, as granitic outcrops were abundant at both sites. Tectonic deformation should not play a significant role, either in the oscillations or in the uniform shift in elevations between the two sites (Methods; Supplementary Figs 3–4; Supplementary Table 6). Last, ponding is unlikely to be a significant factor amongst the fossil corals in our study: the 80-km separation between the Belitung sites (Fig. 1) requires a remarkable coincidence to explain the similar oscillations at the sites if these oscillations were primarily artifacts of localized ponding at each site.

**Comparison with distal records.** The Belitung RSL record is the highest resolution in the mid-Holocene yet obtained in East or Southeast Asia. Only one previous study from the region[17] resolves centennial-scale submetre fluctuations in RSL before 6,000 cal years BP. Interestingly, that data set—a RSL history from southern China based on the surveyed elevations of the upper surfaces of coral microatolls—tells a story of similar rapid oscillations. We reinterpreted the published RSL curve, considering not only the upper surfaces of the microatolls but also the coral diedowns (Methods; Supplementary Table 7). The RSL curves from southern China and Belitung are plotted together in Fig. 9 and all suggest a peak in RSL ~6,800 cal years BP, followed by a trough in RSL ~0.6 m lower, and then a second RSL peak ~6,590 cal years BP, ~0.2 m lower than the first. Additional minor fluctuations at the southern China site, with an

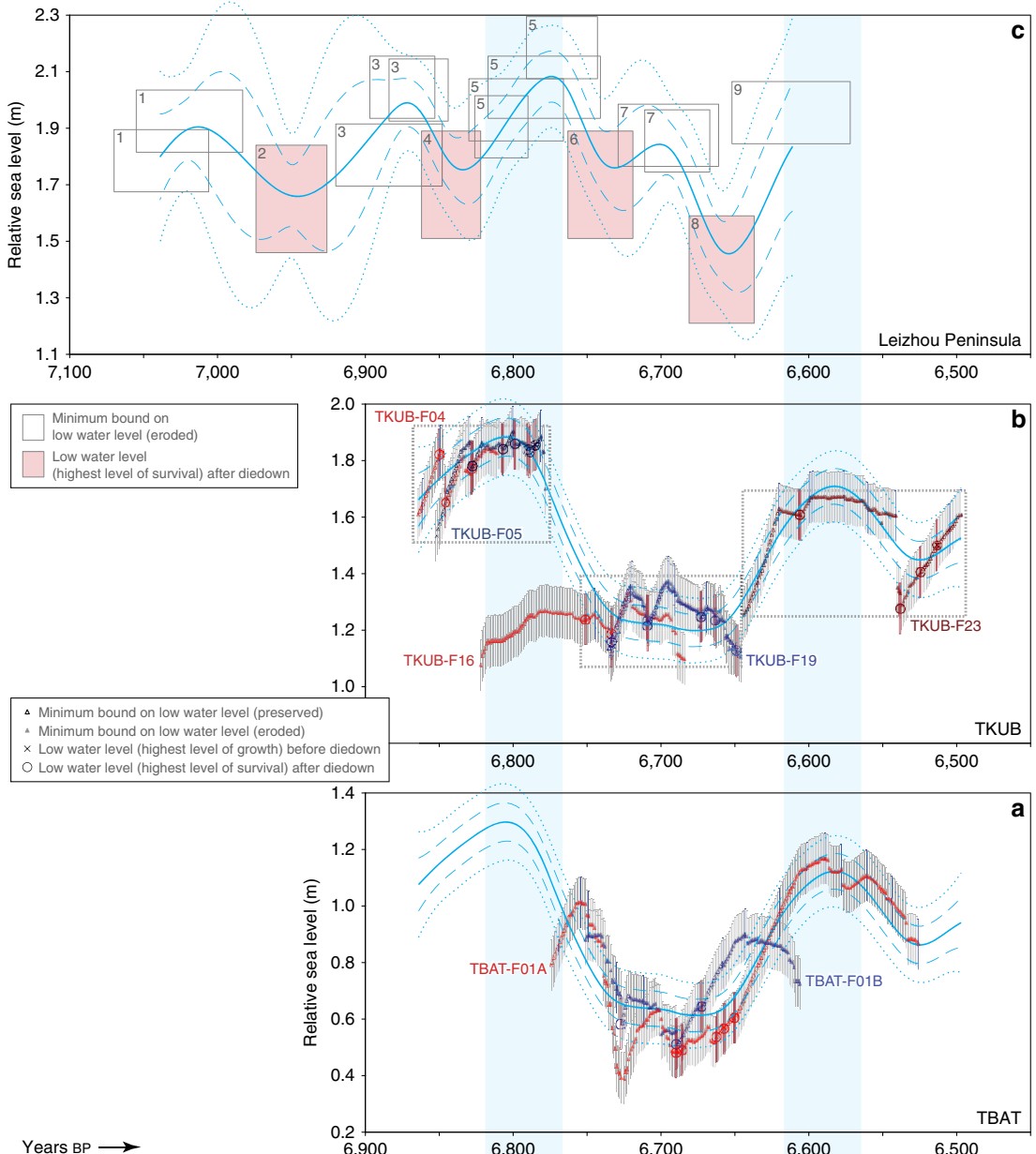

**Figure 9 | Modelled RSL histories.** Posterior estimates of mid-Holocene RSL from the optimized model for the three sites: (**a**) TBAT; (**b**) TKUB; and (**c**) Leizhou Peninsula[17]. For TKUB, the timing of each discrete floating chronology (indicated by a dotted box) is shifted from that shown in Fig. 8, based on model optimization. Unmodelled uncertainty in $\Delta R$ could allow the results for sites TKUB and TBAT to be shifted uniformly by several additional decades, allowing for a slightly improved fit (cyan bars) between the RSL histories at those sites and at the Leizhou Peninsula site. At Leizhou Peninsula, microatoll morphologies allow us to place groupings of data points (numbered 1–9 in the upper left corner of each box) in chronological sequence; this sequence was imposed on the reported U-Th ages (Supplementary Fig. 10), and the ages were then refined using Bayesian modelling techniques[43]. Amplitude hyperparameters for the Leizhou Peninsula site were scaled by a factor of 2 compared with those at the TKUB and TBAT sites, to compensate for poorer data quality at Leizhou Peninsula. Data show $\pm 1\sigma$ vertical and $\pm 2\sigma$ chronological uncertainties; dashed and dotted model curves depict $\pm 1\sigma$ and $\pm 2\sigma$ error envelopes.

intermediate peak ∼6,700 cal years BP, may reflect additional complexity in ocean circulation that has a more profound impact north of Belitung Island.

## Discussion

The similarities between the RSL curves from Belitung Island on the Sunda Shelf and from the southern coast of China, 2,600 km to the north, suggest that the records reflect widespread changes in sea level. To put the ∼0.6 m mid-Holocene fluctuations in

context, annual mean sea level in some modern tide-gauge records is seen to change by as much as 0.2–0.3 m on interannual timescales[2], and the interannual s.d. of sea surface height between 1979 and 2013 approached 0.1 m in some portions of the western Pacific[7]. Using coupled climate models, Widlansky *et al.*[7] project a 5–25% increase in the interannual standard deviation over most of that region for 2006–2100. Many of the regions of high sea-level variability were also areas of extraordinarily high rates of sea-level rise (approaching 30 mm per year) between 1993 and 2001 (ref. 2), though those high rates have been shown to be

biased by the aliasing of interannual and decadal variability into linear sea-level trends over the brief period of observation[3,4]. Although the highest 1993–2001 rates are higher than those inferred from the mid-Holocene corals, the mid-Holocene rates were averaged over and sustained for considerably longer periods of time. Indeed, the amplitude of the mid-Holocene fluctuations on the Sunda Shelf and in the South China Sea exceeds any observed there in modern times. On the Great Barrier Reef in Australia, reconstructions of centennial scale ≥0.3-m RSL fluctuations 5,500–5,100 years BP[30], and 4,800–4,500 and 3,000–2,700 cal years BP[31] suggest that oscillations may be more common than previously appreciated, particularly in the tropics, but sufficiently high-resolution RSL proxy records are needed to identify them. If a similar oscillation were to occur in East and Southeast Asia in the next two centuries, it could directly impact tens of millions of people and associated infrastructure. If this oscillation were to begin with a RSL fall, it would (in the short term) mitigate regional effects of projected eustatic sea-level rise. But if it were to begin with a pronounced RSL rise, this would occur on top of, and exacerbate the effects of, projected long-term global sea-level trends.

The observed RSL fluctuations may result from changes in dynamic sea surface height, local steric effects or eustatic changes. The Southeast Asia–Northern Australia region has considerable interannual and decadal sea-level variability associated with phenomena such as ENSO, the Pacific Decadal Oscillation and the Asian–Australian monsoon[2–4,32]. Over the 17-year period from 1993 to 2009, ENSO and Pacific Decadal Oscillation-related signals raised sea-level trends by 4–6 mm per year north of Australia and by up to 12 mm per year in the tropical western Pacific[3,4]. Effects of interannual and decadal climate variability on the Sunda Shelf and South China Sea have been smaller, but still significant (∼2 mm per year), since 1993 (ref. 3). If either of these climate oscillations entered a persistent strengthened or weakened state over sufficiently long timescales during the mid-Holocene, it is conceivable that they contributed to the sea-level fluctuations observed on Belitung and in southern China, through either dynamic or steric changes in sea level. A coral-based proxy record of tropical Pacific climate variability over the past 1,100 years[33] reveals variations in the strength and frequency of ENSO activity at multi-decadal to centennial timescales—suggesting that variability in ENSO at relevant timescales is physically possible—but the amplitude of sea-level variability in Southeast Asia that would result from such climate fluctuations is unknown. Alternatively, the sea-level fluctuations documented in our study might have been triggered by a shift of the Inter-Tropical Convergence Zone, which would affect the strength of the monsoon. Today, sea-level extremes in the South China Sea (up to ±0.25 m) are primarily monsoon driven[34], but it is unclear how this would be different under a stronger or weaker monsoon. Unfortunately, the poorer temporal resolution of existing regional paleoclimate proxy data from the mid-Holocene limits our ability to make meaningful comparisons (Supplementary Note 1; Supplementary Fig. 5). If the 0.6-m amplitude fluctuations within decades are a global signal, then they imply a heretofore-unknown instability in the mid-Holocene global ice budget. Beyond Southeast Asia, existing ice and sea-level records do not have the necessary resolution to test such a hypothesis[35,36], and models are equivocal as to whether such fluctuations are permissible[37]. High-resolution RSL proxy records from other tectonically stable sites in Southeast Asia, and records spanning more recent time periods, would permit a better understanding of the spatial scale of these sea-level oscillations and could provide insight into whether the period from 6,850 to 6,500 cal years BP was unique.

## Methods

**Distribution of HLG and HLS elevations.** During visits to the TKUB site in 2010, 2011 and 2013, and a visit to the TBAT site in 2013, we surveyed HLG elevations on multiple living coral microatolls. On each coral, we surveyed multiple HLG points, following Meltzner and Woodroffe[15], and we calculated an average HLG elevation for each microatoll. Some of the microatolls appeared to be connected to the open ocean, whereas others were clearly in ponded settings; hence, our distribution of living HLG elevations includes both open-ocean and ponded microatolls. For those corals that appeared in the field to be open-ocean, we checked their locations on high-resolution imagery to ensure that we did not fail to recognize potential moating, which can be difficult to recognize in the field when water levels are higher. In most cases, the imagery confirmed that the surveyed microatolls were seaward of any potential ponds on the reef, but in a few cases, the microatoll's setting could not be unambiguously determined. We therefore classified each microatoll as either clearly open-ocean, clearly ponded or possibly ponded.

We wish to determine the indicative meaning of coral HLS immediately after a diedown, as this is the most direct measurement of RSL, and it is the parameter most easily measured in fossil microatoll slabs. What we were able to survey, however, is coral HLG, a number of years after the most recent diedown. Based on a slab through an unponded living microatoll at TKUB[38], we determined that the most recent diedown occurred in 2005, coincident with the lowest predicted tides of the 18.61-year cycle, and that the microatolls would have grown up 0.06–0.10 m between then and our surveys in 2010–2013. We therefore subtracted 0.06 m from all HLG elevations surveyed in 2010, 0.07 m from all HLG elevations surveyed in 2011 and 0.10 m from all HLG elevations surveyed in 2013. (The upward growth rate tends to be slower in the first few years after a diedown, so the corrections are not exactly proportional to the time elapsed since the most recent diedown as shown simplistically in Fig. 2.) The spread of coral elevations in Fig. 3 represents the distribution of elevations of HLS immediately following a diedown. HLG elevations in subsequent years would be higher than the elevations shown, by an amount dependent upon the coral growth rate and the time since the most recent diedown.

**Calculation of tidal datums.** To determine the indicative meaning of coral HLS, we must determine the range of coral HLS elevations relative to tidal datums at each site. To calculate tidal datums, we used the Oregon State University regional tidal inversion for the Indian Ocean region[39,40]. We extracted the harmonic constituents for each site and used them to calculate mean high water and mean low water (MLW) using formulas from the Manual of Harmonic Constant Reductions[41]. We note that, because the Belitung region is characterized by diurnal tides, mean high water is equivalent to mean higher high water and MLW is equivalent to mean lower low water (MLLW) at each site. We also determined highest astronomical tide and lowest astronomical tide (LAT) for each site by first computing predicted tide levels every hour over an 18.61-year tidal cycle, and then finding the maximum and minimum elevations. The tidal datums are shown in Fig. 3; note the substantially larger tidal range to the northwest.

**Distribution of HLS elevations relative to tidal datums.** To tie the surveyed coral elevations into the tidal cycle, we deployed a portable tide-gauge apparatus (a pressure sensor water-level datalogger, with an accompanying barometric pressure datalogger to barometrically correct data recorded by the water-level datalogger) for just over 5 days at TKUB on northwestern Belitung and for just over 2 days at TBAT on southeastern Belitung. Water-level readings were recorded every 10 s and then smoothed to 1-min intervals. We surveyed the base of the water-level datalogger relative to the corals, and we also periodically (several times per day) surveyed the actual water elevation in a calm (but unponded) area, to validate the water-level datalogger readings.

We extracted tidal predictions from the regional tidal inversion for the Indian Ocean region[39,40]. We adjusted these predictions for local sea-level anomalies estimated on a daily basis from satellite altimetry by the AVISO (Archiving, Validation and Interpretation of Satellite Oceanographic data) group[42]. We then plotted the tide-gauge readings (and surveyed water elevations) against the adjusted tidal predictions, and we uniformly shifted the vertical reference frame of the entire survey (including the water-level readings and the coral elevations) to minimize the misfit between the recorded water levels and the adjusted tidal predictions. This placed all surveyed corals into a vertical reference frame relative to mean sea level. The resulting coral HLS elevations are plotted alongside the various tidal datums for each site on Fig. 3.

**Microatolls at the TBAT site.** At the TBAT site on southeastern Belitung, we found a population of microatolls, spread over a minimum distance of 200 m, with each microatoll at a similar elevation as one another. Each of these microatolls was characterized by a high central dome that was surrounded by low middle concentric annuli and high outer annuli (for example, Fig. 4). The morphology of these corals requires growth under changing RSL conditions. The central dome of each microatoll grew during a period when RSL was high; RSL then fell rapidly, killing the upper portions of the corals; RSL then stabilized at a lower elevation, forming a series of low concentric annuli ∼0.6 m higher than present-day

analogues; RSL then rose ~0.6 m in less than a century, allowing the coral to grow upward to 1.2 m higher than modern living corals.

The biggest and best preserved of this population of microatolls (TBAT-F01) had a radius of 4 m, although a significant sector of the coral had irregularities in its outermost annuli, and the radius in that portion of the microatoll was slightly shorter. The outermost part of the microatoll had also cracked and broken away from the central dome, a consequence of the precarious morphology that resulted from its growth pattern, though it was not difficult to fit the inner and outer parts back together. Nonetheless, because of the cracking and irregularities in the outermost annuli, we extracted two radial slabs from this microatoll, and we used each slab to redundantly constrain the site's RSL history.

For each slab, X-rays were processed and mosaicked together following the guidelines of Meltzner and Woodroffe[15]. In particular, the brightness and contrast of the X-rays were adjusted to emphasize annual banding, but care was taken to avoid introducing artifacts in the final photomosaics, particularly at the boundary between individual X-rays. The banding visible in the photomosaics was then traced, and the RSL history recorded by the coral was interpreted. The annotated slabs (with photomosaics removed for clarity) are shown in Fig. 5. Full-resolution X-ray photomosaics of each slab, and all original unmodified X-rays, are available from the corresponding author.

**Radiocarbon analysis at the TBAT site.** A total of eight radiocarbon samples were dated from TBAT-F01. The radiocarbon dates were modelled using the OxCal calibration program[43]. We applied the Marine13 radiocarbon age calibration curve[22], assuming the marine reservoir correction $\Delta R \approx +89$ years, based on a $\Delta R$ value established from an early 20th century sample from southwestern Borneo[23,44]. Although there is considerable uncertainty in any $\Delta R$ value and its extrapolation spatially and to samples from the mid-Holocene, we can establish that, whatever $\Delta R$ was at our sites at the time, it did not vary in a statistically significant way over the lifetimes of our mid-Holocene corals. This observation is crucial, because it allows us to ignore uncertainties in $\Delta R$ if we are concerned with only the relative age, or the difference in age, between two corals at the same site.

We use the following argument to demonstrate that $\Delta R$ at TBAT did not vary over time. Comparing the unmodelled calibrated radiocarbon dates, assuming for now that $\Delta R \approx +89$ years (with zero uncertainty about that assumed value) and accounting for the number of annual growth bands separating the various samples, seven of the eight ages agree at $1\sigma$ and all agree at $2\sigma$ (Supplementary Table 2). This is consistent with the hypothesis that the reported laboratory errors and the calibration curve correctly describe the uncertainty: 68% of data should agree at $1\sigma$, and 95% should agree at $2\sigma$. This agreement precludes significant variation in $\Delta R$ over the 250-year lifetime of TBAT-F01; if the marine reservoir correction varied by more than a few decades over that period, we would not expect such consistency among the unmodelled radiocarbon dates.

**Microatolls at the TKUB site.** At the TKUB site on northwestern Belitung, no single coral recorded the complete RSL history from ~6,750 to ~6,550 cal years BP, but we compiled a RSL history for the period 6,800 to 6,440 cal years BP from five individual microatolls that all grew over a 3-km stretch. Slabs from each of these corals, TKUB-F04, TKUB-F05, TKUB-F16, TKUB-F19 and TKUB-F23, are shown in Fig. 7.

**Radiocarbon analysis at the TKUB site.** At least two radiocarbon samples were dated from each TKUB coral, and all dates are consistent with their counterparts from the same coral colony at $1\sigma$ (Supplementary Table 2). Although $\Delta R$ at TKUB may differ from $\Delta R$ at TBAT, the consistency among the unmodelled TKUB dates precludes significant variation in $\Delta R$ over the lifetime of each coral at the TKUB site.

**RSL reconstruction at the TKUB site.** The coral growth history based on the TKUB slabs, plotted in Fig. 8, can be divided into three discrete floating chronologies that have been independently radiocarbon dated, but those dates preclude a combined gap of more than ~21 years between the floating chronologies. These three floating chronologies have been merged together in sequence based on the dating results. TKUB-F04 and TKUB-F05 are the oldest and highest microatolls, and they constitute the oldest floating chronology. They grew at a similar elevation as one another and overlapped in time. TKUB-F16 also started growing at about the same time, but it was ~0.6 m lower than TKUB-F04 and TKUB-F05. For at least 70 years, it grew with no indication of a diedown or of even being close to HLS. Within two decades after TKUB-F04 and TKUB-F05 died entirely, presumably from sea-level fall, TKUB-F16 recorded its first diedown; it recorded a second diedown, lower than the first, 18 years later. TKUB-F19 began growing at about this time and recorded its first diedown when TKUB-F16 recorded its second diedown. TKUB-F16 and TKUB-F19 continued to grow and to track RSL for nearly a century, forming the middle floating chronology. TKUB-F23, the youngest of the five corals, forms the third floating chronology. Its elevation and growth history suggest that, within two decades after the death of TKUB-F19, RSL rose rapidly, up to a peak only ~0.2 m lower than the earlier peak recorded by TKUB-F04 and TKUB-F05. More than a century after TKUB-F23 began growing, RSL fell

rapidly over less than a decade or two, then gradually rose again over the following ~30 years.

**Types of observations from microatoll slabs.** We distinguish four types of observations from a coral slab: uneroded HLS elevations immediately following a diedown; uneroded HLG elevations immediately before a diedown; uneroded HLG elevations in years during which no diedown occurred, when the coral was in unrestricted upgrowth mode; and eroded HLG elevations (the highest level of preserved coral growth) for which it is unknown whether a diedown occurred. The first data type (HLS) is the most direct measurement of RSL, but it tracks only the most extreme low tides and may be biased by an unusual climate or weather event that results in a short duration lowering of sea level. The other data types (HLG) are all technically minimum bounds on low water level, because their elevations are controlled by the coral growth rate and not by RSL. The second data type (HLG just before a diedown) is considered to be a closer approximation to RSL than the third and fourth data types, but such data points are rarely preserved[14].

**Vertical uncertainty.** We distinguish two types of vertical uncertainty in our study. The first is aleatoric and quantifiable: random errors that affect the elevation of one part of a curve relative to another part of the same curve. This accounts for the natural distribution of HLS elevations in any population of corals, including the possible effects of unrecognized ponding. Ponding is a phenomenon whereby some corals can survive at higher elevations than they could otherwise, in elevated enclosed pools that do not drain fully at low tide[15,45,46]. Ponding is not always easy to recognize, as the effect can be gradual: one pool may raise the water level at extreme low tide by only a few centimetres over the level in an adjacent pool immediately seaward. Nonetheless, the cumulative effect of multiple subtle ponds at progressively higher elevations tends to exceed 0.1 m only on the wider and more physiographically complex reefs[46].

To estimate a formal uncertainty about the elevation of any one RSL proxy data point, we surveyed a distribution of HLG elevations on living corals (including some that were clearly ponded) at each site. We augmented this data set with the elevation differences between coeval diedowns seen in slabs from two different living corals at the TKUB site[38]. The s.d. of differences in elevation of coeval HLG or HLS at each of our Belitung sites is 0.090 m. This is consistent with observations in Australia, but slightly larger than estimates from off the west coast of Sumatra[15]. The wider distribution of coral HLS on Belitung than off the west coast of Sumatra may occur because of the wider reefs on Belitung, and/or because the tidal range there is larger.

Because ponding is a concern in sea-level studies using coral microatolls, we specifically address whether our results might be biased by ponding in ways that we have not yet considered. At the TKUB site, because the RSL curve was constructed from five separate corals, it is possible that some of the higher and more landward corals (TKUB-F04, TKUB-F05 and/or TKUB-F23) were ponded by significant amounts, that is, by >0.1 m. However, the amplitude of the mid-Holocene oscillations is twice the range of HLS observed among living microatolls on the modern reef, even considering the highest ponded corals (Fig. 3). At the TBAT site, ponding is less likely to explain the observed oscillations, as the oscillations are entirely recorded on individual microatolls. Finally, the two sites are located 80 km apart, on opposite sides of Belitung Island (Fig. 1). This separation is sufficient that it would require a remarkable coincidence to explain the similar changes at the two sites if those changes were caused primarily by localized ponding at each site.

The second type of vertical uncertainty is epistemic and affects the elevation of the entire RSL curve as a whole. These systematic vertical errors are not shown on any figures in this paper, but include uncertainty in the change in tidal range at each site; uncertainties in tectonic effects or compaction at each site; and uncertainty in the HLS elevation of living corals at each site, which is used as the reference elevation for past RSL[15,47]. These errors are difficult to quantify, but they are likely small. Tide modelling (Supplementary Fig. 2) and tectonic modelling (Supplementary Figs 3–4) suggest both of those effects are on the scale of centimetres, and neither compaction of the thin sediments underlying the fossil corals nor ponding of the living microatolls is likely to bias the RSL curve at a site by more than ~0.1 m.

**Sea-level statistical model.** To analyse the RSL proxy data, we constructed an empirical hierarchical statistical model[25], separated into three levels: a data level, which models the recording of RSL by proxies; a process level, which models RSL at the different sites; and a hyperparameter level, which characterizes key attributes of the first two levels.

At the data level, RSL index points (HLS elevations following diedowns) from Belitung are preserved typically once or twice per 18.61-year nodal tidal cycle, whereas minimum limiting data (HLG elevations, or minimum bounds on low water level) are resolved each year. We use all of the index points, as they are indicative of sea level. The selection of limiting data is more complicated, however, as our model treats limiting data as faithful sea-level indicators, yet in reality some limiting data are severe underestimates of sea level. Specifically, any limiting data from before a microatoll's initial diedown represent coral growth up to that initial HLS, and these data may be decimetres (or even metres) below HLS[14]. Even after a coral's initial diedown, due to patterns of microatoll growth over the 18.61-year

tidal cycle (Fig. 2), some limiting data from our sites are expected to be as much as 0.20 m lower than the theoretical HLS; in these cases, the highest limiting data point within each 18.61-year cycle should be a reasonable approximation of theoretical HLS for that year, and therefore a useful proxy for RSL. In principle, erosion should also be considered at the data level, but because we selected slabs that were well preserved, erosion was negligible (~0.05 m or less) and can be ignored over most of the time series in our study. An exception to this is the later RSL peak at both sites, ca. 6,600–6,550 years BP, where no diedowns are preserved and erosion may locally exceed 0.15 m. Because of this limitation, our model may underestimate the elevation of the second RSL peak, and the amplitude of the fluctuations we infer in our study should be considered a conservative minimum estimate.

Our preferred strategy for modelling limiting data from the Belitung sites is, therefore, to subsample the limiting data by selecting only the highest limiting point in each 18.61-year bin (Supplementary Fig. 6); nonetheless, we also consider an alternative strategy, in which we use the highest limiting point available for each year (the only point available in most years), excluding only the early part of TKUB-F16, before the coral had grown up to HLS (Supplementary Fig. 7). The preferred strategy is an attempt to use only data that reliably approximate a given year's theoretical HLS; the alternative strategy is an attempt to use as much of the limiting data as is possibly justifiable.

We model noisy proxy observations ($y_i$) of RSL elevation as

$$y_i = f_j(t_i + \Delta_k) + \varepsilon_i \qquad (1)$$

where $i$ indexes data points and $j$ indexes sites, and the function $f_j(t)$ is RSL at site $j$ and time $t$. Each observation belongs to one of four floating chronologies (the entire record at TBAT, plus three discrete floating chronologies at TKUB), indexed by $k \in [0, 3]$; each floating chronology is associated with an age shift $\Delta_k$. The sea-level observation errors, $\varepsilon_i$, are treated as uncorrelated and normally distributed, with a s.d. of 0.09 m determined as discussed in the text and earlier in Methods.

Coral ages are constrained by radiocarbon dating methods. Because we can assume that the marine radiocarbon reservoir correction, $\Delta R$, is fixed over time at each site, the relative age uncertainties between the three floating chronologies at the TKUB site are determined by the radiocarbon ages presented in Supplementary Table 2; these inter-slab age uncertainties result in the possibility that one, two or all three of the TKUB floating chronologies are as much as 21 years older. In addition, uncertainty in $\Delta R$ at each site allows for an inter-site relative age shift between the overall time series at the TKUB site and that at the TBAT site of up to approximately $\pm 120$ years (the $\pm 85$-year uncertainty from each site added together in quadrature). Because the modelling depends only upon relative ages and not upon absolute ages, and because the inter-site relative age uncertainty is so much larger than the intra-site relative age uncertainties, we need only three age-shift parameters, $\{\Delta_0, \Delta_1, \Delta_2\}$, and we can define them in a way that is more intuitive than elicited by the formula above (we can fix $\Delta_3$ at 0 year). For convenience, we hold the time series at TBAT fixed to that determined assuming $\Delta R = +89$ years, as discussed in the text. $\Delta_0$ is the overall age shift of the TKUB record relative to the TBAT record, and we allow $-120$ years $\leq \Delta_0 \leq +120$ years. $\Delta_1$ and $\Delta_2$ are the age shifts of the oldest and youngest floating chronologies at the TKUB site relative to the central floating chronology at the site, such that the sum of $\Delta_1$ and $\Delta_2$ is a maximum of 21 years (and a minimum of 0 year), where $\Delta_1$ and $\Delta_2$ are shifts in opposite directions, $\Delta_1$ making the oldest slabs older and $\Delta_2$ making the youngest slab younger. Age uncertainties within individual floating chronologies are not incorporated into the model, as the law of superposition prohibits swapping the order of data derived from successive annual bands, effectively rendering the relative age uncertainty to be negligible.

At the process level, $f_j(t)$ is specified as the sum of a common (shared) regional sea-level signal $g(t)$, a periodic signal representing the 18.61-year nodal tidal cycle $p_j(t)$, a site-specific offset $c_j$, and high-frequency variability $w_j(t)$:

$$f_j(t) = g(t) + p_j(t) + c_j + w_j(t). \qquad (2)$$

The prior distribution of the shared signal, $g(t)$, is a mean-zero Gaussian process[48] characterized by hyperparameters that comprise an amplitude $\sigma_g$ and a timescale of variability $\tau$,

$$g(t) \sim GP\left\{0, \sigma_g^2 \rho(t,t';\tau)\right\}, \qquad (3)$$

where $\rho$ is the Matérn correlation function with smoothness parameter $^3/_2$ and scale $\tau$. The use of a smoothness parameter of $^3/_2$ ensures that the first derivative of the process will be defined everywhere, but allows for abrupt changes in rate.

The prior distribution of the periodic signal representing coral growth over the nodal tidal cycle, $p_j(t)$, is a mean-zero GP characterized by hyperparameters that comprise an amplitude $\sigma_p$, a smoothness parameter $v_p$ and a fixed period corresponding to the nodal tidal period, 18.61 years[49]:

$$p_j(t) \sim GP\left\{0, \sigma_p^2 \exp\left(\frac{-2\sin^2\left(\frac{\pi(t-t')}{18.61}\right)}{v_p^2}\right)\right\}, \qquad (4)$$

where $t$ and $t'$ are defined in years. The hyperparameters of this periodic component[48] are tuned for each site to simulations of coral growth under present-day nodal tidal cycles at the site. We assumed a coral growth rate $r$ that is normally

distributed with a mean of 12 mm per year and s.d. of 2 mm per year and a periodic cycle with tidal amplitudes ($\sigma_p$) of 0.186 and 0.089 m at TKUB and TBAT, respectively (Fig. 2). For tuning these hyperparameters, simulated RSL is given by:

$$RSL(t) = \sigma_p^2\left(1 + \cos\left(-\pi + \frac{2\pi t}{18.61}\right)\right). \qquad (5)$$

The simulated coral height at any given time, CH($t$), is equal to the minimum of RSL($t$) and the potential growth of the coral according to the randomized growth rate, based on the coral height in the previous year, CH($t-1$) + $r$:

$$CH(t) = \min(CH(t-1) + r, RSL(t)). \qquad (6)$$

We generate five random, 100-year-long time series at each site and fit these synthetic coral height data to a mean-zero GP, equivalent to the periodic component of the process model plus white noise. We use these maximum-likelihood parameters from this exercise as the amplitude and smoothness hyperparameters in $p_j(t)$ of the original process level, above.

The prior distribution of the constant site-specific offset, $c_j$, is normal with mean zero and variance $\sigma_c^2$. We restrict this site offset to being constant because we do not expect any physical processes to give rise to significant centennial-scale or sub-centennial-scale variations in RSL between the two sites. The two sites should be exposed to essentially indistinguishable dynamic sea-level changes, and any tectonic deformation at these sites should be small and similar at the two sites (discussed later in Methods). While GIA and changes in tidal range do vary spatially, any changes due to these processes should be small enough on a centennial scale that they are well within any noise.

The prior distribution of the high-frequency variability in RSL, $w_j(t)$, is modelled as white noise, with a normal distribution with mean zero and variance $\sigma_w^2$, and its additional homoscedastic (equal variability) noise.

We employ an empirical Bayesian analysis method, in which the age-shift parameters $\{\Delta_0, \Delta_1, \Delta_2\}$ and the hyperparameters $\{\sigma_g, \tau, \sigma_c, \sigma_w\}$ are point estimates calibrated based on the data to maximize the likelihood of the model. The hyperparameters $\{\sigma_p, v_p\}$ are optimized as described above, based on the present-day tidal cycles and coral growth models at TKUB and TBAT, and are held constant during the optimization of the other hyperparameters. The key output of the model is an estimate of the posterior probability distribution of the RSL field, $f_j(t)$, conditional on the tuned hyperparameters (Supplementary Figs 6–9; Supplementary Table 3).

In the end, the model based on our preferred strategy does a reasonable job of separating the non-linear and periodic signals (Supplementary Fig. 6), and the rates of RSL change it estimates should reflect secular trends, minimally biased by vagaries of coral growth variability over the 18.61-year tidal cycle. The alternative model, in contrast, does a poor job of separating out the periodic term, and it forces more high-frequency variability into the non-linear signal, likely overestimating short-term rates of sea-level change. Although we suspect that the high-frequency variability (period ~30 years) seen only in the alternative model (Supplementary Fig. 7c) is an artifact of that model trying to fit limiting data that severely underestimate theoretical HLS, the fact that both strategies yield fluctuations at a 200-year timescale with peak-to-trough amplitudes of 0.5–0.7 m and similar timing suggests that these model results are robust.

**Reinterpretation of published data from southern China.** Yu et al.[17] surveyed, sampled and dated a suite of coral microatolls from a site on the Leizhou Peninsula, along the southern coast of China. Unlike in our study, where we collected and analysed full radial slabs of each microatoll, they presented primarily point data from the upper surfaces of microatoll annuli. In total, they published 13 dated samples, each of which was tied to the elevation from which it was collected. They also provided photos and cross-sectional sketches of each microatoll, so although those authors focused only on the upper surfaces, they provided enough information to estimate the timing and elevations of the more prominent diedowns.

We reinterpreted the RSL curve of Yu et al.[17] (Supplementary Fig. 10) by estimating the timing and elevations of those more prominent diedowns. The reported ages were based on U-Th techniques (typically with small errors) and were all in the expected sequence (ages from the outer annuli of each micro-atoll were sequentially younger than ages from the inner annuli), so it was straightforward to estimate the timing of each diedown, and to correlate diedowns from one coral to another. Numerous points in each photograph were marked with surveyed elevations, providing a sense of scale, so we were able to estimate the elevations of those diedowns with sufficiently conservative vertical errors.

Last, we correlated coeval annuli from one coral to another based on their reported ages. Again, this was straightforward, as the microatolls provide a consistent, reproducible RSL history, with the same number of prominent diedowns on the various microatolls between any two dates. The advantage of correlating the annuli manifests itself when considering the handful of U-Th ages in their study that had sizable errors. In the few cases where the chronological errors were so large that the sample age overlapped with sample ages from adjacent annuli, our effort to group the age–elevation data based on the coral morphologies allowed us to minimize the ambiguity of whether a particular sample belonged on one downward swing of the RSL curve or on the subsequent upward swing (Supplementary Fig. 10).

The most notable difference between the Yu et al.[17] interpretation and our reinterpretation of the Leizhou Peninsula data is that, in our analysis, the amplitudes of the RSL fluctuations have increased. To give credit to the original authors, Yu et al.[17] acknowledged that their RSL curve 'only represents minimum cycles of fluctuation, because the low-lying gouges were not dated' and that 'the amplitudes of sea-level fluctuations should also be treated as representing minimum values'. Yu et al.[17] had observed fluctuations based simply upon the differential heights of the sequential annuli (although they missed the third RSL drop, which is required by the third diedown on their microatoll-3), and adding the HLS (diedown) elevations to their RSL curve causes the troughs of the curve to drop lower. The lowering of the RSL troughs is robust even allowing for 0.1–0.2 m coral growth over the 18.61-year nodal tidal cycle, as we have done for our Belitung time series.

We conservatively increase the vertical errors reported by Yu et al.[17] to match those determined for the Belitung sites, and we arbitrarily double the vertical errors for the diedowns, since they are estimated from photos. Furthermore, unlike the Belitung time series where data points were annually resolved and relative age uncertainties were negligible, here there are only several limiting data points per century, each has errors spanning two 18.61-year cycles or more, and only the most prominent diedowns (one per century) have been identified. Hence, it is not practical to isolate the 18.61-year periodic component of the time series from the Leizhou Peninsula. Instead, we add additional vertical error (in quadrature) to account for the fact that this signal is not modelled. The reinterpreted data set, with the increased vertical errors and added diedowns, appears in Supplementary Table 7 and Supplementary Fig. 10. Note that Yu et al.[17] had dated two samples (FPO-26 and FPO-30) from the same annulus of the same microatoll at the same elevation, and the samples provided similar ages; we combine them here and derive a weighted average of the ages.

**Modification of the sea-level model for southern China data.** Because the large chronological errors on a few data points in the Leizhou Peninsula data set allow for the swapping of the ordering of those points in ways that are clearly impossible (based on coral morphologies and the law of superposition), it is desirable to trim the chronological errors where appropriate. We ran the original U-Th ages in Supplementary Table 7 through OxCal[43], classifying each grouping of data as an unordered Phase() in OxCal; each of these nine phases was separated from adjacent phases by a Boundary(), thereby imposing the sequential ordering of diedowns that is known from the coral morphologies. The OxCal-trimmed ages, which were used in subsequent analyses, appear in Supplementary Table 7 under the heading 'modelled ages'.

As noted earlier, the Leizhou Peninsula data comprise only a few data points per century, each with errors spanning two 18.61-year cycles or more, rendering it unjustifiable to model the periodic tidal cycle signal; hence, the Leizhou Peninsula model includes only a non-linear signal and a white noise term. The non-linear signal is based on the data from the higher-resolution Belitung sites: it retains the optimized timescale derived from the Belitung sites, but we scale the amplitude hyperparameters by a factor of two compared with the Belitung sites, to compensate for the fact that with the sparser data set, the model with the unscaled amplitude hyperparameters systematically under-predicts the RSL peaks and over-predicts the RSL troughs. An additional complication is that the Leizhou Peninsula data set contains 12 minimum limiting data points (which are clustered near and constrain the peaks of the RSL curve) but only four index points (which are isolated at and constrain the troughs of the RSL curve). An unweighted model tends to fit the limiting data at the expense of fitting the index points; this is effectively a sampling bias. To compensate, we triple the weight on each index point, so that the index points as a whole (the data constraining the troughs) and the limiting data as a whole (the data constraining the peaks) are weighted equally. This has the desired effect that the model fits the peaks and the troughs of the RSL curve more equitably. The model is plotted against data with their original U-Th dates in Supplementary Fig. 10 (this figure also shows the effect of scaling the amplitude hyperparameters), and it is plotted against data with model-refined ages in Fig. 9.

**Sea-level model cross-validation.** Cross-validation is used to compare the performance of different predictive modelling procedures. For the preferred model (Supplementary Fig. 6), we performed an exhaustive (64 runs, one for each training point) Leave-One-Out Cross-Validation of the model. Since the model is tuned to envelop 95% of the data, we expect the point that is left out of the optimization of the model to be included ∼95% of the time. Supplementary Table 8 shows the number and percentage of data points that were within the 95% interval of our model's posterior predictive distribution. The model achieved 92.2% inclusion within the prediction interval. In addition, Supplementary Table 8 shows the mean, median and median absolute value of all of the differences (or residuals) between predicted RSL and sea-level height of the data point. For the mean and median, over-predictions and under-predictions tend to cancel one another out, so values near zero suggest that the differences are randomly distributed. For a model that treats each training point as a sea-level index point, we expect such behaviour. The median absolute error is the median of the absolute value of each difference, so values near zero suggest better predictive power of the model.

**Differential GIA across Belitung.** Modelling GIA is a complex problem and has been the focus of much research in recent decades. There remain large uncertainties both in spatio-temporal details of the ice melting history[50] and in the most appropriate rheological model (lithosphere thickness, and upper and lower mantle viscosities) to use. Significant differences persist in the values of these parameters assumed by different global GIA models[51–54]. However, a recently developed GIA model for the Southeast Asia region consistently shows that RSL over the past 7 kyr should be higher at site TKUB on northwestern Belitung than at site TBAT on southeastern Belitung, regardless of the choice of earth model and ice model[28].

At far-field sites, following ice melting and the inundation of broad, shallow continental shelves, the RSL signal is driven primarily by two GIA processes: equatorial ocean syphoning and continental levering[27]. Equatorial ocean syphoning results in far-field RSL fall, due to the migration of water from the far field to the near field to fill the regions vacated by the collapsing forebulge. Continental levering from increased ocean load along continental margins induces uplift in inland regions and subsidence within the ocean basin, generating large sea-level gradients perpendicular to the coast (as shown on Supplementary Fig. 1a). It is the interplay between these two processes and the spatially complex signal resulting from the two larger nearby landmasses of Sumatra (to the west) and Borneo (to the east) that drive the difference in the RSL signal between TKUB and TBAT.

In a bathymetrically simple region, such as off the south coast of China, features formed at sea level (such as a wave-cut notch, abrasion platform, or coral reef) during the mid-Holocene on a small island far offshore would now be below sea level, even if sea level itself has not risen since the mid-Holocene. In contrast, similar sea-level markers formed inland (such as in an embayment) would now be higher than when they formed. In the narrow region between Sumatra and Borneo, the GIA signals from these two larger islands interact; this interaction drives a differential uplift signal across Belitung Island (for example, Supplementary Fig. 1a), with TKUB subjected to a larger ocean-load driven RSL fall than TBAT.

Supplementary Fig. 1b shows a $\chi^2$ analysis of various rheological models used to predict RSL, from Bradley et al.[28]. Several potential models are listed in Supplementary Table 5. The '96p28' model, which yields the best fit overall to Holocene data from the Malay–Thai Peninsula, predicts that the RSL at 7 kyr should have been 0.23 m greater at TKUB than at TBAT (Supplementary Table 5 and Supplementary Fig. 1a–d). The '9611' model, which yields only a slightly poorer fit to the Malay–Thai data but produces a marginally better fit to Holocene data from China (but which is still outside the 95% confidence limit for the preferred earth model for China), predicts that the RSL at 7 kyr should have been 0.40 m greater at TKUB than at TBAT (Supplementary Table 5; Supplementary Fig. 1b–d). The '96p510' model would be considered a global-average earth model, and although it falls outside the 95% confidence limit for the preferred earth model for the Malay–Thai Peninsula, it would predict that the RSL at 7 kyr should have been 0.39 m greater at TKUB than at TBAT (Supplementary Table 5; Supplementary Fig. 1b). We note that the earth models that fit the China data well and those that fit the Malay–Thailand data well constitute two generally distinct populations of models. This might be expected, given the significantly different tectonic regimes across the two regions. Although details remain to be resolved and efforts to do so are an active research area, a consistent conclusion from the range of plausible models considered by Bradley et al.[28] is that the RSL at 7 kyr was decimetres higher at TKUB than at TBAT. This is supported further by the model of Peltier[51], which incorporates the ICE-5G (VM2) ice model–earth model combination and predicts that RSL at 7 kyr was 0.38 m higher at TKUB than at TBAT.

We therefore conclude that much of the 0.5–0.7 m discrepancy between the absolute elevations observed for mid-Holocene RSL at TKUB on northwestern Belitung and TBAT on southeastern Belitung can be explained and is predicted by GIA.

**Tidal range across Belitung and its change over time.** Belitung Island sits at an exceptional location on a map of tides. As shown on Fig. 3, the tidal range is substantially larger at site TKUB than only 80 km to the southeast at site TBAT. Supplementary Fig. 2a, which shows the spatial distribution of the elevation of MLLW relative to mean sea level, reveals a tight gradient in tidal amplitudes across the island. Previous modelling studies[55–57] have demonstrated that this gradient is primarily due to spatial differences in the amplitudes of diurnal tidal constituents ($K_1$ and so on). The variability of the $K_1$ constituent is a shelf-resonance; the length and width of the basin produce a natural period of oscillation that is closely aligned with the period of the $K_1$ tide. We wondered whether the resonance pattern or the tidal range at either site might have been different during the mid-Holocene, when local RSL was 1–2 m higher. This is important because HLS tracks lower water levels, and any change over time in the tidal range could bias our reconstructions of RSL during the mid-Holocene.

Answering this question required the application of a dynamical tidal model rather than assimilative model such as TPXO7.2 (ref. 58). Therefore, the two-dimensional depth-integrated version of ADCIRC[59] was applied to the region. A large area (Supplementary Fig. 2d) was modelled in order to place the model open boundaries in regions of deep water (Indian Ocean and Pacific Ocean), where small depth changes would not be expected to affect the tidal constituents. ADCIRC uses an unstructured triangular mesh, and mesh resolution was adjusted

to place the highest resolution (kilometre scale) in the Java Sea. The final mesh had ~1 million elements. Mesh bathymetry was drawn from a blend of SRTM30 data[60] and global ASTER data. Initial attempts to use ETOPO1 bathymetry[61] revealed strongly non-physical artifacts in our study area.

The open boundaries of the model domain were forced with tidal constituents drawn from the FES2004 tidal model[62]. For the model runs representing higher RSL, the FES2004 constituents were again used. The inherent assumption is that small changes in water depth in the deep ocean locations of the open boundaries will not affect the tides. This assumption is supported by Hill et al.[63], who found that large changes in water depth could markedly affect regional or shelf tides (for example, in the western North Atlantic Ocean) but that metre-scale changes had little impact in deep basins. To compute tidal datums, a model run of 90 days was conducted, with the first 30 days as a ramp-up period. Harmonic analysis was performed on the remaining 60 days, yielding amplitudes and phases for each tidal constituent within the domain. These amplitudes and phases were subsequently converted to tidal datums (mean higher high water, MLW and so on) using the Harmonic Constant Datum method[64].

Supplementary Fig. 2a–b shows the results for MLLW for the baseline condition and the condition where the water depth has been uniformly increased by 2 m. As discussed earlier, the geometry of the Java Sea is resonant with the $K_1$ tide and the northwest–southeast gradients in MLLW are essentially a proxy for the spatial variability in the $K_1$ tide (and, to a lesser extent, in the other diurnal constituents).

Supplementary Fig. 2c shows the change over time of the MLLW elevation. For both sites, the model predicts that MLLW would have been several centimetres lower under the conditions of ~6.5 kyr, but the change would have been larger at TBAT. For LAT, the changes over time would have been approximately twice those for MLLW: at ~6.5 kyr, LAT would have been 0.00–0.05 m lower at TKUB on northwestern Belitung and 0.05–0.10 m lower at TBAT on southeastern Belitung.

Two types of modelling artifacts show up in the contour maps (Supplementary Fig. 2a–c) and should be ignored. The occasional crescent 'artifacts' observed in the contour maps are due to the region being at a boundary between predominantly semi-diurnal and predominantly diurnal basins. This boundary is quantified by the amplitude ratio R[64]. The Harmonic Constant Datum method uses slightly different techniques for diurnal and semi-diurnal regions and these slight differences are responsible for the crescent features in Supplementary Fig. 2a–c. In a basin that is strongly semi-diurnal, or strongly diurnal, these artifacts would be absent. In addition, the baseline scenario (Supplementary Fig. 2a) has elements that 'dry out' at extremely low tide. The tidal signals at these nodes have truncated troughs, which produce unreliable estimates of amplitudes and phases from the harmonic analysis. These results then propagate into the calculation of tidal datums, and they propagate further into the map of the change over time; the strongly blue bits of Supplementary Fig. 2c are likely an artifact of MLLW being incorrect in those areas in Supplementary Fig. 2a.

These modelling results, taken in consideration of the observed fossil coral elevations, imply that, during the mid-Holocene, both mean sea level and LAT were higher than today, but there was a greater separation between mean sea level and LAT. If the corals directly indicate that LAT at ~6.5 kyr was at the elevations plotted in Figs 6 and 8 (compared with LAT today), then mean sea level at ~6.5 kyr would have been 0.00–0.05 m higher at TKUB and 0.05–0.10 m higher at TBAT (compared with mean sea level today) than shown on those plots. These differences are small compared with other errors and can be ignored for most purposes, but we note that a small fraction (perhaps 0.05–0.10 m) of the 0.5–0.7 m discrepancy between the apparent elevations observed for mid-Holocene RSL at TKUB on northwestern Belitung and TBAT on southeastern Belitung can be explained and is predicted by changes in the tidal range over time.

**Inferred tectonic stability of Belitung from GPS data.** Although Belitung Island is considered tectonically stable, few data exist with which to test any hypothesis of regional tectonic stability. Simons et al.[16] published and analysed GPS data from >100 sites across the Southeast Asia region, spanning 10 years, from 1994 to 2004. They defined a relatively undeforming Sundaland block and characterized its boundaries (Fig. 1). Their results suggest that Belitung Island is tectonically stable, at least over the period of their study. While some studies suggest that geodetic deformation can vary on multi-decadal timescales and that 10 years of data are far from sufficient to understand deformation on timescales of centuries to millennia, at least above the seismogenic zone of a subduction megathrust[20], there are no mapped active tectonic faults near Belitung, and no evidence has yet been recognized that suggests the Belitung vicinity has been tectonically active over the Holocene.

**Potential viscoelastic response to megathrust rupture.** The suggestions of tectonic stability in recent decades notwithstanding, Belitung is only 700 km from the Sunda–Java trench, and places such as Phuket, Thailand, at a similar distance from the 2004 rupture, have experienced substantial vertical deformation as part of a viscoelastic response to the 2004 earthquake[65]. We therefore modelled the potential effects on Belitung of two scenario ruptures along the Sunda megathrust.

We developed models in VISCO1D, using two end-member rheologies previously determined for the Sunda megathrust, to predict the viscoelastic response following a hypothetical rupture along the portion of the Sunda–Java megathrust closest to Belitung Island. The first end-member rheology is that of

Pollitz et al.[66,67]; the second end-member rheology is that of Panet et al.[68] Both were constrained by postseismic observations following the 2004 and 2005 Sunda megathrust ruptures, but using different data sets. Parameters of each rheological model are given in Supplementary Table 6.

For each end-member rheology, we modelled the response to two hypothetical ruptures. In each scenario, we assumed the fault ruptures the megathrust up to the surface. We placed the hypothetical ruptures along portions of the Sunda–Java megathrust stretching from southern Sumatra to western Java, which would maximize deformation at Belitung. Although these specific ruptures are neither known nor expected from historical or geological information, the seismogenic potential of this section of the megathrust is poorly understood, and we wished to consider 'worst-case' plausible scenarios. The first scenario earthquake for each rheology has $M_W$ 8.9, with rupture dimensions of 518 by 175 km and uniform slip of 11.0 m (Supplementary Fig. 3a,e). A rupture of this size in this location may already be pushing or exceeding the limits of what is possible along this section of the megathrust. The second scenario earthquake for each rheology has $M_W$ 9.2, such as in 2004, but is more compact, with rupture dimensions of 748 by 236 km and uniform slip of 14.9 m (Supplementary Fig. 3b,f). Rupture dimensions for the chosen magnitudes are based on the scaling relations of Blaser et al.[69]. The rake angle in both scenarios is 90°, that is, pure thrust motion.

For the rheology of Pollitz et al.[66,67], the predicted gradual viscoelastic response at Belitung to a large megathrust rupture centred about the Sunda Strait is trenchward and downward (Supplementary Fig. 3a–d). For the $M_W$ 8.9 earthquake, the cumulative vertical displacement at Belitung after 50 years is ~0.10 m, which is substantially smaller than the amplitude of the oscillations in RSL recorded by the corals. Even for the $M_W$ 9.2 rupture, the vertical displacement at Belitung after 50 years is only ~0.26 m, which is larger but still less than half the amplitude of the observed RSL oscillations.

In contrast, using the rheology of Panet et al.[68], the model predicts a gradual viscoelastic response at Belitung that is trenchward and upward (Supplementary Fig. 3e–h). For the $M_W$ 8.9 earthquake, the cumulative vertical displacement at Belitung after 50 years is also ~0.10 m, which again is substantially smaller than the amplitude of the oscillations in RSL recorded by the corals. Even for the $M_W$ 9.2 rupture, the predicted vertical displacement at the TKUB and TBAT sites after 50 years is ~0.28 m or less, still less than half the amplitude of the observed RSL oscillations.

In principle, if the magnitude of viscoelastic deformation were larger (in either direction), it might explain a single oscillation at Belitung. If the postseismic viscoelastic response were downward, then if RSL had reached its peak on the Sunda Shelf and had already begun to fall when, ~6,750 cal years BP, a large rupture occurred near the Sunda Strait, the response at Belitung over the following 50 years or more would have been one of land-level fall and RSL rise. Once this postseismic viscoelastic response decayed to a negligible rate, RSL fall due to GIA would have resumed, and the RSL highstand at Belitung would have been characterized by a double peak. Alternatively, if the postseismic viscoelastic response were upward, then if the large rupture occurred roughly 150 years before RSL would have otherwise reached its peak due to GIA, then the ongoing sea-level rise would have been overwhelmed by postseismic uplift, and RSL would have begun falling. Once this postseismic viscoelastic response decayed to a negligible rate, RSL rise dominated by the meltwater signal would have resumed, until the RSL highstand, and then after the meltwater production ceased, GIA processes would have led to RSL fall; again, the apparent RSL highstand at Belitung would have been characterized by a double peak.

Nevertheless, we find each of these explanations unsatisfying and unable to explain the totality of the observations for at least four reasons: it would require a coincidence in the timing of the earthquake, either only a few decades after the peak in RSL (in the first scenario) or about 150 years before the peak (in the second scenario); it cannot explain more than two RSL peaks, whereas corals indicate at least three occurred; it cannot explain coeval oscillations 2,600 km away along the south coast of China[17] (Figs 1 and 9); and as discussed above, the amplitudes of a predicted viscoelastic response to coseismic rupture simply are not large enough unless we invoke an unrealistically large coseismic rupture to trigger the response or a rheology very different from those that have been published. As a caveat to this last point, we note that the rheologies assumed for the postseismic models (Supplementary Table 6) and the rheologies assumed for the GIA models (Supplementary Table 5) are quite different. The incompatibilities between the postseismic rheologies and the GIA rheologies are common in such studies. This serves, in part, to illustrate how poorly rheology models are constrained. In any case, the four reasons above are sufficient to discount the likelihood that viscoelastic deformation following coseismic rupture along the Sunda–Java megathrust played any role in our observed oscillations on Belitung.

**Potential deformation from an unknown upper-plate fault.** The 04 June 2015 $M_W$ 6.0 Sabah earthquake, 1,400 km northeast of Belitung, occurred on a fault that was previously unrecognized, highlighting the dearth of understanding of the tectonics of this intraplate region. In the hope of considering all possible tectonic explanations for the observed RSL oscillations at Belitung, we also explore hypothetical deformation that might occur if something like the 2015 Sabah rupture occurred on an unknown fault closer to Belitung. As in the previous subsection, we seek merely to answer the question of whether two of the RSL peaks at the two

Belitung sites might be explained by coseismic rupture of an upper-plate fault. Specifically, what minimum moment magnitude would be needed for an optimally located and optimally oriented upper-plate rupture to generate ~0.6 m of uplift or ~0.6 m of subsidence simultaneously at both Belitung sites?

Using a dip angle similar to the 2015 Sabah rupture (70°) and the scaling relations of Blaser et al.[69], we generated a series of synthetic ruptures up to $M_W$ 7.6, each with a depth of slip that allows the rupture to propagate to the surface (Supplementary Fig. 4). The smallest rupture that produces uplift or subsidence of ≥0.6 m at a single site has $M_W$ 6.8 (Supplementary Fig. 4a); however, the vertical deformation signal is localized over an area with a maximum dimension of ~30 km, precluding ≥0.6 m of vertical deformation simultaneously at two sites 80 km apart. The smallest rupture that could produce uplift or subsidence of ≥0.6 m simultaneously at two sites 80 km apart has $M_W$ 7.6 (Supplementary Fig. 4e); a fault capable of such an earthquake would necessarily be longer than 80 km and have obvious geomorphic expression. In contrast, if any active upper-plate fault exists near Belitung, it must be short enough, with little enough cumulative slip, to have thus far gone unnoticed by geologists. We therefore conclude that rupture of an upper-plate fault is not a viable explanation for the RSL oscillations observed on Belitung between 6,800 and 6,500 cal years BP.

**Potential deformation from deeper earthquakes.** We discount the likelihood of vertical deformation near Belitung, on the order of ≥0.6 m, from an intermediate-depth or deep-focus earthquake. Although moderate ruptures at depths of ~600 km or more occurred in 1957 ($M_W$ 7.2) and 1963 ($M_W$ 7.1) only ~200 km south of Belitung, and although a larger ($M_W$ 7.5) intermediate-depth event occurred in 2007, 350 km to the south[70], it is generally observed that intermediate-depth and deep-focus ruptures occur within the subducted slab and tend to produce only centimeter-scale deformation at the surface.

**Data availability.** Data and modelling codes that have contributed to the reported results are available from the corresponding author upon request.

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

## Acknowledgements

Primary funding for this work came from the National Research Foundation Singapore and the Singapore Ministry of Education under the Research Centres of Excellence initiative. This research was also supported by the National Research Foundation Singapore under its Singapore NRF Fellowship scheme (Awards NRF-RF2010-04 and NRF-NRFF2010-064), and by Academic Research Fund (AcRF) Complexity Tier 1 Project RGC4/14. R.E.K., E.A. and B.P.H. were supported by NSF (ARC-1203415 and OCE-1458904). S.L.B. acknowledges support from the Netherlands Earth System Science Centre (NESSC), which is financially supported by the Ministry of Education, Culture and Science (OCW). We thank A.D. Hawkes, D. Prayudi and I. Suprihanto for field support; A. Ritch, G. Lee and B. Jonnalagadda for slab analysis; D.T. Pham and D. Warrick for tide analysis; J. Pile and K.E. Bradley for GIS assistance; M. Sim and C. Prior for discussions on radiocarbon dating; K. Sieh for initial discussions; and A. Thomas for reviews. This paper results from joint-research activities between the Earth Observatory of Singapore at Nanyang Technological University and the Research Center for Geotechnology LIPI. This paper is a contribution to PALSEA2 (Paleo-Constraints on Sea-Level Rise 2) and to International Geoscience Programme (IGCP) Project 639, 'Sea Level Change from Minutes to Millennia'. This is Earth Observatory of Singapore contribution 120.

## Author contributions

A.J.M. designed and oversaw all aspects of the research, led field work and took the lead on writing the manuscript. A.D.S. and B.P.H. guided the intellectual direction of the research. E.A. and R.E.K. developed and applied the hierarchical sea-level models. Q.Q. and E.M.H. modelled potential tectonic deformation at our sites. D.F.H. modelled changes in tidal range over time. S.L.B. modelled effects of GIA. A.D.S., B.P.H., J.M.M., D.H.N. and B.W.S. assisted with field work. Selected portions of the manuscript or supplement were written by A.D.S., B.P.H., E.A., Q.Q., D.F.H. and S.L.B. All authors reviewed the manuscript.

## Additional information

**Competing financial interests:** The authors declare no competing financial interests.

