## [Peer Review File · Nature Communications]

Reviewers' comments:

Reviewer #1 (Remarks to the Author):

The authors present a detailed study of coral microatolls from southeast Asia, which they use to reconstruct relative sea-level changes for a 300-yr period within the mid-Holocene. The paper is well-written and will be of interest to a wide audience because it quantifies relatively rapid pre-industrial sea-level changes in a low-lying and (now) densely populated region, where to date there are few high-resolution sea-level reconstructions. It is suitable for publication in Nature Communications, but with some major structural revision plus a consideration of the effects of dynamic topography.

Main concerns

While the authors have thoroughly accounted for the quantitative effects of various processes on their sea-level reconstructions (GIA, tectonic, tidal amplitude, methodological uncertainties), they have not considered steric effects or changes in dynamic (ie, ocean surface) topography. Both can be significant on the timescales considered here. A global/regional satellite sea-level map would be a nice inclusion in the paper, to place the authors' observations in context. In that sense, the title and overall message of the paper may be misleading - yes the sea-level oscillations are relatively large, but they are in the same range as modern satellite observations for the region, as well as sea-surface topographic variations in general. That is not to say that this study isn't important - it provides much-needed high-resolution sea-level data for an interglacial period, and the authors are very thorough in their approach, but dynamic topography needs to be more fully considered and quantified, and the results discussed in that context.

My other main issue is that the manuscript is hard to follow as it stands, simply due to its structure. The main text is too brief while the Supplementary Information and Methods are excessively long. The Methods can be shortened so that they are more concise and don't include background information. And there is key information in the SI/Methods that would be better placed in the main text. There is also a lot of repetition between the main-text/SI/Methods; this adds to the manuscript length and disrupts its flow. The main text therefore could and should be extended to include parts from the SI/Methods. There is also room for a 4th figure in the main text. Also, the main text doesn't follow the standard Nature Communications format of a 'Results' section (with sub-headings) and a final 'Discussion'.

Statistics

Overall these are robustly dealt with, especially the statistical sea-level model. However, given this nice model, the representation of the individual site RSL histories is rather simplistic by comparison. Referring to main-text Fig. 2 and Supp. Figs S9, S10: the grey shading is misleading - it looks like an envelope for credible intervals when it is actually based on assuming a uniform uncertainty about a hand-drawn curve (black line in figures). Why wasn't a more probabilistic approach taken for the individual TBAT and TKUB RSL histories? I'm still not entirely clear about how the nodes were chosen in the black curve, or how the 1 and 2 σ about the black curve was calculated. The coloured symbols in Figs S9 & S10 should show error bars (plus the symbols themselves are too small). And regarding the vertical uncertainties, the tidal range uncertainty is ~ 15 cm (is that ± 1 or 2σ ?), so shouldn't this be combined with the 9 cm (1s.d.) HLS? And what about adding an uncertainty for ponding? Supp Figs S9 & S10 could maybe go in the main text as 1 figure.

Other points

Palaeoclimate context (Supp Note S7, Fig. S16): These are rather naïve and do not provide any useful insight, so I would drop them (the manuscript won't suffer). In Fig. S16, different palaeoclimate proxy records are plotted together in order to try to offer a regional climatic context. These include cave $\delta^{18}\text{O}$ from a wide geographic range of sites, as well as a δD leaf wax record. Notwithstanding differences between δD and $\delta^{18}\text{O}$, cave $\delta^{18}\text{O}$ alone is affected by various

processes (eg $\delta^{18}\text{O}$ of source water & precipitation, rainfall amount), which can lead to very different cave $\delta^{18}\text{O}$ signals from even proximal sites. There is no/little coherence among the δD and $\delta^{18}\text{O}$ records shown in Fig. S16, and the authors' RSL time series is far too short compared to the other records for a meaningful comparison of trends. The fact that this is a monsoon region adds further complexity to proxy and climate interpretations, and we simply cannot link the sea-level reconstructions to these climate records with any certainty (in fairness, the authors acknowledge the limitations of this exercise, but I would just drop it completely).

Line 57-58: the die-downs to which the authors refer are not explicitly indicated on Supp Figs S3-S10

Line 134-136 (& Fig. 2a): the authors state they reinterpreted a previous RSL curve, but give insufficient detail about how exactly they did this (nor are the original & reinterpreted data given in a data table).

Line 157-158: insert 'within decades' after oscillations and 'Holocene' before global ice budget

Figures S3-S8: text on figures (especially black text) is too small

Figures S4-S8 could be fit onto 1-2 pages as 1 figure (parts a,b,c, etc)

Figures S9: red text 'dotten' should be 'dotted'.

Review of ‘Large regional sea-level oscillations on human timescales, revealed by mid-Holocene corals,’ by Meltzner et al.

This article addresses the problem of determining the occurrence and oscillatory behaviour of relative sea level (RSL) from data extracted from coral microatolls on Belitung Island in Indonesia. Establishing RSL fluctuations from coral microatolls is a challenging problem, due to the erosion of the atolls, the possibility of confounding lowering RSL with ponding or extreme tides in observed diedowns, and due to the inherent inaccuracies in radiocarbon dating. The authors acknowledge all these inherent problems and use a hierarchical statistical model to capture all sources of variability and uncertainty. The authors conclude that their data corroborates that of Yu et al. (2009), in that the RSL oscillated in the period in question (spanning around two centuries) with an amplitude of around 0.6 m. The authors do not speculate as to whether the change was local, global or an isolated incident.

There is much to like about this article and I appreciate the effort the authors went through to make it readable to non-specialists. I will comment on the article from a statistical perspective. Gaussian process modelling for this problem is a perfect fit and, I believe, innovative. There are some aspects of the modelling and the presentation, however, that could do with some improving, and I am slightly concerned with the validity of the cross-validation analysis. This is an important part of a nice work and crucial to get right. I hope the authors find my assessment and the suggestions below helpful.

The statistical analysis

The data model:

- The authors subsampled the limiting data points, so as to “avoid weighting the model more heavily with the much more abundant limiting data.” This, however, is not a valid reason, especially in an application where data is so costly to obtain (except for cross-validation, but see later point). Further, artificially censoring data is rarely needed: When data are not ‘trusted’ then they can be weighted differently (i.e., they generally are associated with a larger uncertainty or with a bias, otherwise why not use them?). On closer inspection, however, the authors have better reason to discard most of the limiting data as they are not direct readings of RSL. In fact, as the authors state several times (e.g., lines 204–222), limiting data is a lower bound on RSL due to the lag between RSL and polyp growth. My interpretation of the parameter α in line 311 is that, provided a consistent set of limiting data is chosen, then one can safely assume that limiting data is always offset from the true RSL by α . Their consistent set is obtained by choosing the highest data point in the 18.6 yr nodal tidal cycle; this choice seems defensible to me. What I find hard to justify, however, is the use of the other limiting data as validation data. The parameter α is estimated on the assumption that HLG data is constantly offset from the HLS data by a fixed amount. While this may hold for all maxima in the tidal cycles, this almost certainly does not hold for all the limiting data which occur at different time points in the tidal cycle. So my first impression is that the cross-validation exercise is flawed.

When validating, it is important that the validating data could plausibly have been generated by the same underlying model that was trained. I believe a way around this would be to explicitly model α as a temporally-varying function. For example, one could replace α with $\alpha \sin(\phi t + \gamma)$, where ϕ is calibrated so that the period of the sin curve is 18.6 years, and γ is either empirically estimated or chosen by looking at the data. Then one may use the limiting data for training as well, and maybe a randomly chosen subset for validation. In any case, this aspect of the model needs looking into as I am not convinced it is representative, and this may lead to erroneous conclusions with cross-validation.

- The measurement error ϵ : If I understand correctly, the authors have defined $\sigma_\epsilon^2 = \sigma_u^2 + \sigma_y^2$, where σ_y^2 is the pooled variance of the data. How is σ_y^2 obtained? I don’t see how this can be obtained without some prior fitting of the trend (i.e., of $f(t)$), otherwise this will be too large. If σ_u^2 is also unknown then a safer approach would be to just estimate σ_ϵ^2 directly. In any case, the components of variability are not placed correctly within the hierarchical model. I presume y (up to an offset) is known with high accuracy, and measurement accuracies associated with it can be fixed to some reasonable value. All observed variability is because of the fine-scale process variance, σ_u^2 , which should really be in the process model (in line 333) and not in the measurement model since, as the authors state, it is used to capture variability of unresolved *sea processes*. This would not considerably change the optimization routines and their conclusions on $g(t)$, but it will affect the posterior uncertainties on $f(t)$. Inference is then still carried out on the RSL $g(t) + k_j$ which is the main component of interest.

The process model:

- The temporal process (RSL) is modelled as a Gaussian process. Since the measurement time point is not exactly known, this is an error-in-variables problem. First, using ϵ for both the error in variables and measurement error is confusing, I would use a different symbol. Second, the authors state that “geochronological uncertainties were not incorporated into the hyperparameter optimization for computational efficiency.” I found this a bit surprising given that there was so much talk about the uncertainty of radio-carbon dating. Clearly there are uncertainties that can be defensibly ignored (e.g., intra-coral-slab measurements) and other that can not (intra-site slabs). The authors use 4 FCAs, but this is in fact a small number given all the ways in which the FCAs at the TKUB site can be shifted, and one may always argue that there are scenarios that weren’t explored. Crucially, considering only a few cases will not yield tenable uncertainty quantification assessments. I suggest the authors explore better ways of incorporating these uncertainties (these would go into ϵ^t) and do away with the use of subjectively chosen FCAs/binning methods. This would require some prior modelling assumption on the relationship between HLS and HLG states (e.g., using the sin curve above) and possibly MCMC, but the resulting inferences will be more defensible.
- The Taylor expansion in line 327 implies that $\text{var}(\epsilon_i^t)$ needs to be specified for each i , yet there is no talk of ϵ_i^t anywhere else in the text. Is it site-specific? How is it used? Was the variance of ϵ_i^t estimated or was it fixed to something reasonable? The authors did not use it for hyper-parameter estimation but one of the benefits of linearisation is that a linear-Gaussian model is recovered, and this should not upset the computational efficiency substantially.
- From the arguments presented in the text, I suspect the authors are not including any systematic component in ϵ_i^t so that they use FCAs instead. However, a more realistic model for ϵ_i^t would be $\epsilon_{i,j}^t = \mu_j + \zeta_i$ where μ_j captures the coral’s specific shift and ζ_i is just white noise. This, again, would allow the authors to do away with the FCAs and to obtain inferences on RSL that capture all of the uncertainty (plus, one would get objective information on the specific shift μ_j). Note that μ_j is in units of years and is different from k_j , which is in mm, is site-specific, and which I imagine should be well-constrained. I am not convinced linearisation is needed for computational efficiency in this problem, even if a systematic component is considered: I suspect McHutchon and Rasmussen (2011) developed this model for when one has multiple input dimensions, of which the authors only have one. I briefly show an example where I try to estimate the systematic component from the data at the end of this section; adding noise with known variance to this systematic component would be straightforward.
- The notation needs some improving. In line 334, replace $g(t_i)$ with $g(t)$. Also in line 327 f should be site-specific (i.e., use f_j), and $s(t)$ on page S.21 is also defined to be site-specific.

The cross-validation:

- The cross-validation exercise is an important one. However I have my concerns with the use of all the limiting data with the proposed model; please see these above. The authors have constructed a score (based on the cross-validated likelihood, or log predictive score) using *all* data points. Using all the data points is non-standard, since one is including training data when assessing model fit. I suspect that using all the data in the scoring will not penalize for over-fitting as much as one would wish. Why not carry out a typical cross-validation exercise and just validate using the left-out data? Also, it would be good to show other diagnostics of the cross-validated data such as the mean absolute prediction error or some hybrid score such as the continuous ranked probability score. If this cross-validation procedure was taken from a book or paper please provide a citation (general note: Please provide more citations for statistical methods employed).
- I found the notation in this section a bit difficult to follow. Sometimes P is used to denote a probability, and sometimes a probability density function. Also the reader is left to guess what many things are. For example, is f_L the trained process evaluated at the limiting data or the process trained with the limiting data? L is a set since the authors use $i \in L$; what is this set? I suggest this section is revised to make the notation clearer.
- I am not sure what the final paragraph in 6.2 is suggesting. In particular, I don’t see how one can validate against the true signal $f(t)$ if this is not known. Since one only has observations y , then the error term needs to be included in the likelihood when validating. White noise contains no information, and will not contribute to predictive performance in the mean. I suspect the authors have made this choice because they are using all the data (including the training data) in the cross-validation. This would not have been an issue if only the validation data was used, as is typically the norm when assessing model fit.

- In Table S5 it is important that the math symbols are used as the labels are subject to interpretation (what is σ with no subscript? Is it a confidence interval? Does ‘Time scale’ refer to τ ? If so why are the units mm and not yr?). Note that if the HLS and HLG relationship is modelled and the shifts are estimated (which I believe is possible, see below) then this table would collapse to one row.

Use of MCMC in estimating intra-site carbon dating errors

I believe the authors can do away with linearity assumptions on the error and should attempt a full analysis without FCAs. I did a quick test to check whether this is indeed possible. Let $x(t)$ be a Gaussian process with a Matérn correlation function with smoothness parameter $3/2$ as defined by the authors. Then assume that $y_1(t) = x(t) + \epsilon_1$ and $y_2(t) = x(t + \Delta) + \epsilon_2$. In their case $y_1(t)$ would be the measurements at the TBAT site and y_2 would be data from one of the slabs at TKUB. For this simple example I assumed that the process parameters of $x(t)$ are known, that $\Delta = 50$, and that the variance of ϵ_1 and ϵ_2 was equal to 1. I probably have used more observations than the authors have available but also threw in substantial measurement error.

In order to infer Δ I used MCMC, where I assumed that Δ was a priori normally distributed centred at zero. I obtained 2000 samples in a few seconds on my machine, and I don’t think the authors have so much data that this will become computationally unfeasible. In Fig. 1 I show the true process $x(t)$ (unobserved), y_1 and y_2 . In Fig. 2 I show the posterior distribution on Δ after removing the first 400 samples for burnin-in. This indicates that it is possible to propagate uncertainties in the carbon dating all the way through. I’m not suggesting this will be as easy to implement in the author’s application, but not properly accounting for these uncertainties may influence the inferences, and this is of concern in error-in-variables problems.

Figure 1: Simulated function from Gaussian process prior (black), simulated unshifted data (blue) and simulated shifted data (red).

Figure 2: MCMC chain (left) and histogram (right) summarizing samples from the posterior distribution of Δ given the two data sets.

Reviewer #3 (Remarks to the Author):

Review of Meltzner et al by Alex Thomas

The authors present a new and substantial dataset of proxy sea level from southeast Asia covering a brief time interval from 6800 to 6500 years BP. These data are highly relevant as they provide insight into the potential for multi-decadal sea level variability during the Holocene. Such variability is currently highly contested and would therefore be of interest to a wide range of researchers interested in both paleo and modern sea level variability.

The data comprise the reconstruction of past sea level marker points from coral microatolls and combined radiocarbon and layer counting chronologies. The data are from two independent sites and qualitatively agrees with existing data from the region adding to the confidence in the results. This is an established method for reconstructing sea level but this study provides new data from a previously understudied region.

I feel that the results are robust but there are some points of clarification required regarding the maximum rates of change and some places where the manuscript could be improved prior to publication.

The authors have done a thorough job in constraining the uncertainties in the elevations of the RSL markers and in constructing the age models. The age models within each coral microatoll are highly precise due to the constraint of layer counting but the absolute ages and ages between sites have relatively large uncertainties due to the inherent uncertainty of marine radiocarbon age determinations. The authors are careful in their interpretation not to interpret the relative ages of the sea level changes at different sites and being at the same time, but the limitations of the chronology should be better accounted for in the main text. The uncertainty in the timings of the changes in RSL between the different samples is consistent with the sea level changes happening at the same time but it could be that there is a real difference in timing. This should be more clearly acknowledged and the implications for the potential mechanisms of regional sea level variability discussed.

The rates of change of sea level seem to be inconsistent between the figure and the text. The text claims rates of rise and fall for 18.1 and -23.3 mm/yr (also units are inconsistent some figures used m/kyr) [lines 126-127], but the rates illustrated in Figure 3 (and in the SI) show much lower maximum rates of around 7 and -20. I can't figure out where this discrepancy arises. But this must be reconciled before publication. Additionally there when combining datasets to produce a single RSL curve with a probabilistic model then the more data sources that are included then there will be a smoothing artefact if there is any uncertainty in the age models between the datasets. The rates calculated are therefore likely an underestimate. It would therefore be nice to see the rates calculated for the individual sites as well as the combined.

Perhaps the most disappointing part of the discussion for me is the interpretation of the changes in regional sea level. There seems to be little critical analysis of the potential causes of the variability. The amplitude of 60-70 cm seems large compared to the variability of ENSO or the monsoon. How much of the observations could be explained by these mechanisms? Additionally does the timescale of the changes to fit with known ENSO variability there are hints of similar timescales in the coral data of Kobb et al 2003. Do the spatial patterns of the amplitudes of the variability match for the three records we have versus that expected for ENSO driven or global ocean volume driven changes?

Figure 2 the grey bands showing the RSL histories at each site are clear but it would be good to include in that the key data which is used to constrain each curve (at least for the new sites. This is shown in the SI but it would make it much clearer for non expert readers how the curves were constructed and their relative merits. The key points are the highest level of survivals which could be made more obvious than they are in the SI figures.

Line 105 which way would the tidal biases bias the relative amplitudes?

Line 107 "Preclude" means to prevent, 'exclude' would be more appropriate.

The title is not strictly correct. For this variability to be "oscillations" requires a defined frequency

and amplitude. Neither of these is strictly confirmed. "Vacillations" would be more correct but somewhat esoteric. "Rapid sea level variability on societally relevant timescales"?

Reviewer #1 (Remarks to the Author):

The authors present a detailed study of coral microatolls from southeast Asia, which they use to reconstruct relative sea-level changes for a 300-yr period within the mid-Holocene. The paper is well-written and will be of interest to a wide audience because it quantifies relatively rapid pre-industrial sea-level changes in a low-lying and (now) densely populated region, where to date there are few high-resolution sea-level reconstructions. It is suitable for publication in Nature Communications, but with some major structural revision plus a consideration of the effects of dynamic topography.

We thank the reviewer for recognizing the importance of this study. We address this reviewer's specific comments, below.

Main concerns

While the authors have thoroughly accounted for the quantitative effects of various processes on their sea-level reconstructions (GIA, tectonic, tidal amplitude, methodological uncertainties), they have not considered steric effects or changes in dynamic (ie, ocean surface) topography. Both can be significant on the timescales considered here. A global/regional satellite sea-level map would be a nice inclusion in the paper, to place the authors' observations in context. In that sense, the title and overall message of the paper may be misleading - yes the sea-level oscillations are relatively large, but they are in the same range as modern satellite observations for the region, as well as sea-surface topographic variations in general. That is not to say that this study isn't important - it provides much-needed high-resolution sea-level data for an interglacial period, and the authors are very thorough in their approach, but dynamic topography needs to be more fully considered and quantified, and the results discussed in that context.

The reviewer raises some good points. In our minds, these sea-level changes were indeed some combination of steric effects and changes in dynamic topography, but we did not say that explicitly in our previous draft. We now make that more explicit in the final paragraph, and (as one possible explanation) we relate dynamic and steric effects to ENSO and PDO. We also now cite published modern observations, to place our mid-Holocene observations in context. We could, as the reviewer suggested, include a copy of the map of Widlansky et al. (their figure 2), but we believe it is sufficient to cite their study instead. (If the Editors still feel it would be advantageous to reproduce or modify the figure of Widlansky et al. in our own work, then we will readily take that advice.) We're unclear why our description of the oscillations as "large" was misleading, but in any case, the title has been changed and that point is no longer applicable. However, contrary to our understanding of the reviewer's statement, the mid-Holocene sea-level oscillations are greater than what is seen in modern satellite observations for the region: although the highest 1993–2001 sea-level rates are higher than those we infer from the mid-Holocene corals, the mid-Holocene rates were sustained for considerably longer periods of time, such that the mid-Holocene amplitudes are larger -- and we think that is important. In any case, we feel that dynamic topography is now more fully considered and quantified, and the results are now discussed in that context.

My other main issue is that the manuscript is hard to follow as it stands, simply due to its structure. The main text is too brief while the Supplementary Information and Methods are excessively long. The Methods can be shortened so that they are more concise and don't include background information. And there is key information in the SI/Methods that would be better placed in the main text. There is also a lot of repetition between the main-text/SI/Methods; this adds to the manuscript length and disrupts its flow. The main text therefore could and should be extended to include parts from the SI/Methods. There is also room for a 4th figure in the main text. Also, the main text doesn't follow the standard Nature Communications format of a 'Results' section (with sub-headings) and a final 'Discussion'.

We concur. This problem arose from our initial attempt to format the manuscript for submission to *Nature*. Now that the manuscript has been transferred to *Nature Communications*, we have taken advantage of the more generous word length allotted to articles. In our revised manuscript, we have moved much of what was in Methods and some of what was in the SI into the main text, eliminating repetitive text wherever appropriate. The revised manuscript adheres to the standard *Nature Communications* format, and following the Guide for Submission, the Methods section is now focused mostly on our statistical modeling methodology. We believe that the main text and Methods are now easier to follow. We have also added a fourth figure in the main text, as discussed later in this letter.

Statistics

Overall these are robustly dealt with, especially the statistical sea-level model. However, given this nice model, the representation of the individual site RSL histories is rather simplistic by comparison. Referring to main-text Fig. 2 and Supp. Figs S9, S10: the grey shading is misleading - it looks like an envelope for credible intervals when it is actually based on assuming a uniform uncertainty about a hand-drawn curve (black line in figures). Why wasn't a more probabilistic approach taken for the individual TBAT and TKUB RSL histories? I'm still not entirely clear about how the nodes were chosen in the black curve, or how the 1 and 2 σ about the black curve was calculated. The coloured symbols in Figs S9 & S10 should show error bars (plus the symbols themselves are too small). And regarding the vertical uncertainties, the tidal range uncertainty is ~ 15 cm (is that ± 1 or 2σ ?), so shouldn't this be combined with the 9 cm (1s.d.) HLS? And what about adding an uncertainty for ponding? Supp Figs S9 & S10 could maybe go in the main text as 1 figure.

We thank the reviewer for the compliments. Following the reviewer's suggestion, we have completely done away with the hand-drawn curves and gray shading that appeared in Fig. 2 and Supp. Figs. S9 and S10 in the previous draft. Gray shading now appears in Supp. Figs. S13 and S14, but here it is exactly what the reviewer states that it should be: an envelope for credible intervals, based on our empirical statistical model. We have increased the symbol sizes to 150% of what they were in the previous draft, but the main challenge is that, the bigger we make the symbols, the more each point obscures its neighbors. Another challenge in plotting the symbols is that we want the HLS points (circles) to stand out more prominently because they provide fundamentally different (and more useful) information; the best way we have found to make the circles stand out is to make them bigger, but that then requires a slight range of symbol sizes. In the end we tried to strike a balance between all of these competing interests, but we're definitely willing to consider further suggestions on the presentation of data in this figure. Also, we still don't show error bars in Figs. 2 and 3 or Supp. Figs S13 and S14—this is also to help improve legibility—but we state in the captions that vertical uncertainties are uniformly ± 0.09 m (1σ) about each data point. In contrast, Fig. 4 (which is focused on the bigger picture—models, interpretations, uncertainties, errors, etc.) does show vertical errors bars. We hope this, too, strikes a balance. Regarding the tidal range, we do not consider the ~ 15 cm value to be an uncertainty for the TKUB and TBAT sites; it is a periodic oscillation that we can and do model for those sites. Only for the Leizhou Peninsula dataset do we consider an extra source of uncertainty that results from this periodic oscillation; there, we consider it an uncertainty because the data are too sparse (the sampling frequency is too low) to model the 18.61-yr periodic signal. For the Leizhou Peninsula dataset, we combine the uncertainties as the reviewer suggests (see Methods). Lastly, the ± 9 cm (1σ) uncertainty already fully accounts for ponding on the modern reef, as the calculation was based on a survey of living corals that included the highest (most ponded) corals we could find; we have tried to make that point more explicitly in the revised manuscript (see also Supp. Fig. S2). The previous Supp. Figs. S9 and S10 have been moved to the main text as suggested, but we have removed the hand-drawn curves as noted earlier.

Other points

Palaeoclimate context (Supp Note S7, Fig. S16): These are rather naïve and do not provide any useful insight, so I would drop them (the manuscript won't suffer). In Fig. S16, different palaeoclimate proxy records are plotted together in order to try to offer a regional climatic context. These include cave $\delta^{18}\text{O}$ from a wide geographic range of sites, as well as a δD leaf wax record. Notwithstanding differences between δD and $\delta^{18}\text{O}$, cave $\delta^{18}\text{O}$ alone is affected by various processes (eg $\delta^{18}\text{O}$ of source water &

precipitation, rainfall amount), which can lead to very different cave $\delta^{18}\text{O}$ signals from even proximal sites. There is no/little coherence among the δD and $\delta^{18}\text{O}$ records shown in Fig. S16, and the authors' RSL time series is far too short compared to the other records for a meaningful comparison of trends. The fact that this is a monsoon region adds further complexity to proxy and climate interpretations, and we simply cannot link the sea-level reconstructions to these climate records with any certainty (in fairness, the authors acknowledge the limitations of this exercise, but I would just drop it completely).

We concur with the reviewer's assessment, and we deliberated over whether to leave this in. In the revised manuscript, we have de-emphasized this point even further, but we left it in for one reason: numerous colleagues of ours didn't appreciate the extent to which the paleoclimate proxies disagreed with one another, and numerous colleagues didn't appreciate how much greater the temporal resolution was of our time series, compared to those of most of the paleoclimate proxy records. When presenting our results, numerous colleagues asked if we had attempted the comparison; it was only after seeing the figure (now Supp. Fig. S10) that those colleagues realized a comparison would not be useful. Therefore, we leave it in because we believe our colleagues would be interested in seeing the comparison, if only to convince themselves it is not useful. We will be happy to remove it outright if the Editors still feel it would be best to do so.

Line 57-58: the diedowns to which the authors refer are not explicitly indicated on Supp Figs S3-S10

We thank the reviewer for pointing that out; that was an oversight. The lines of text cited by the reviewer no longer exist, but we have nonetheless labeled the diedowns as "Unconformity (diedown)" on what is now Supp. Figs. S4 and S5. Diedowns are also indicated on Figs. 2-4.

Line 134-136 (& Fig. 2a): the authors state they reinterpreted a previous RSL curve, but give insufficient detail about how exactly they did this (nor are the original & reinterpreted data given in a data table).

We have taken this suggestion to heart, and in the revised manuscript we thoroughly explain what we did, and how our reinterpretation differs from the authors' original interpretation. The second sub-section of the Methods (5 paragraphs!) is now devoted to explaining what we did differently than the original authors, and the effects those differences had on the interpreted RSL curve. The third sub-section of the Methods (2 additional paragraphs!) summarizes a new statistical model (a modification of the empirical model we used for the TKUB and TBAT sites) we have introduced since the previous version of the manuscript; the aim is to make our treatment of the Leizhou Peninsula data more rigorous than in our previous manuscript. The original and reinterpreted data are now presented in Supp. Table S7 and shown on Supp. Fig. S15 and Fig. 4c.

Line 157-158: insert 'within decades' after oscillations and 'Holocene' before global ice budget

We added "within decades" and "mid-Holocene" in the places suggested. We thank the reviewer for the suggestion.

Figures S3-S8: text on figures (especially black text) is too small

We have enlarged all the text, especially the black text, on what is now Supp. Figs. S4 and S5. We ask the Editors to advise us if any of the text is still too small.

Figures S4-S8 could be fit onto 1-2 pages as 1 figure (parts a,b,c, etc)

We have combined former Supp Figs. S4-S8 and the new Supp. Fig. S5 (a-f).

Figures S9: red text 'dotten' should be 'dotted'.

We thank the reviewer for catching that mistake! It has been fixed on what is now Fig. 3.

Reviewer #2 (Remarks to the Author):

This article addresses the problem of determining the occurrence and oscillatory behaviour of relative sea level (RSL) from data extracted from coral microatolls on Belitung Island in Indonesia. Establishing RSL fluctuations from coral microatolls is a challenging problem, due to the erosion of the atolls, the possibility of confounding lowering RSL with ponding or extreme tides in observed diedowns, and due to the inherent inaccuracies in radiocarbon dating. The authors acknowledge all these inherent problems and use a hierarchical statistical model to capture all sources of variability and uncertainty. The authors conclude that their data corroborates that of Yu et al. (2009), in that the RSL oscillated in the period in question (spanning around two centuries) with an amplitude of around 0.6 m. The authors do not speculate as to whether the change was local, global or an isolated incident.

We thank the reviewer for succinctly explaining our work. We'd like to point out that we now speculate that the changes were "at least regional in scope."

There is much to like about this article and I appreciate the effort the authors went through to make it readable to non-specialists. I will comment on the article from a statistical perspective. Gaussian process modelling for this problem is a perfect fit and, I believe, innovative. There are some aspects of the modelling and the presentation, however, that could do with some improving, and I am slightly concerned with the validity of the cross-validation analysis. This is an important part of a nice work and crucial to get right. I hope the authors find my assessment and the suggestions below helpful.

We thank the reviewer again for pointing these things out, and for the suggestions. As outlined below, we feel we have made improvements that address this reviewer's concerns, and the cross validation that we now employ is completely different (and much simpler and more standardized) than previously.

The statistical analysis

The data model:

- The authors subsampled the limiting data points, so as to "avoid weighting the model more heavily with the much more abundant limiting data." This, however, is not a valid reason, especially in an application where data is so costly to obtain (except for cross-validation, but see later point). Further, artificially censoring data is rarely needed: When data are not 'trusted' then they can be weighted differently (i.e., they generally are associated with a larger uncertainty or with a bias, otherwise why not use them?). On closer inspection, however, the authors have better reason to discard most of the limiting data as they are not direct readings of RSL. In fact, as the authors state several times (e.g., lines 204–222), limiting data is a lower bound on RSL due to the lag between RSL and polyp growth.

As the reviewer noted, we have good reason to discard most of the limiting data, but we did a poor job of making our case in the text; we have now addressed that shortcoming. We now explain more carefully and correctly that the problem in data selection is that some limiting data are severe underestimates of RSL, yet the hierarchal model treats all data as sea-level index points.

My interpretation of the parameter α in line 311 is that, provided a consistent set of limiting data is chosen, then one can safely assume that limiting data is always offset from the true RSL by α . Their consistent set is obtained by choosing the highest data point in the 18.6 yr nodal tidal cycle; this choice seems defensible to me. What I find hard to justify, however, is the use of the other limiting data as validation data. The parameter α is estimated on the assumption that HLG data is constantly offset from the HLS data by a fixed amount. While this may hold for all maxima in the tidal cycles, this almost certainly does not hold for all the limiting data which occur at different time points in the tidal cycle. So my first impression is that the cross-validation exercise is flawed. When validating, it is important that the validating data could plausibly have been generated by the same underlying model that was trained. I believe a way around this would be to explicitly model α as a temporally-varying function. For example,

one could replace α with $\alpha \sin(\phi t + \gamma)$, where ϕ is calibrated so that the period of the sin curve is 18.6 years, and γ is either empirically estimated or chosen by looking at the data. Then one may use the limiting data for training as well, and maybe a randomly chosen subset for validation. In any case, this aspect of the model needs looking into as I am not convinced it is representative, and this may lead to erroneous conclusions with cross-validation.

We have taken the reviewer's suggestion to include a periodic sinusoidal-type function within our model. However, because the tidal cycle exhibits strong periodicity but not perfect sinusoidality, we instead added a periodic term to the covariance function. We did this by adding a sinusoidal prior covariance to the model. Specifically, we first produced synthetic coral growth data, simulating growth over sequential nodal tidal cycles (with different amplitudes at each site) as well as random coral growth with reasonable variation. Taking five samples for each site gave us a realistic set of coral data under ideal (synthetic) conditions. Using these synthetic data, we ran a GP regression, optimizing our hyperparameters for maximum likelihood for our sinusoidal prior covariance function. We ended with hyperparameters that fit our synthetic data at both sites well. Therefore, we used this covariance function as our prior distribution of the periodic covariance (with fixed hyperparameters) in our model of the noisier (real) coral data. Details are provided in the Methods.

Also, to investigate whether our conclusions are sensitive to the details of how we subsampled the limiting data, we now include an alternative model in which we use most of the limiting data (Supp. Fig. S12). As expected, this model introduces substantial high-frequency variability that we believe is an artifact of using limiting data that severely underestimate RSL (again, see discussion in Methods). However, the fact that this alternative model retains the 0.5–0.7 m fluctuations at a 200-yr timescale suggests that these results are robust and not sensitive to the details of how we subsampled the limiting data.

- The measurement error ϵ : If I understand correctly, the authors have defined $\sigma_\epsilon^2 = \sigma_u^2 + \sigma_v^2$, where σ_v^2 is the pooled variance of the data. How is σ_v^2 obtained? I don't see how this can be obtained without some prior fitting of the trend (i.e., of $f(t)$), otherwise this will be too large. If σ_u^2 is also unknown then a safer approach would be to just estimate σ_ϵ^2 directly. In any case, the components of variability are not placed correctly within the hierarchical model. I presume γ (up to an offset) is known with high accuracy, and measurement accuracies associated with it can be fixed to some reasonable value. All observed variability is because of the fine-scale process variance, σ_u^2 , which should really be in the process model (in line 333) and not in the measurement model since, as the authors state, it is used to capture variability of unresolved sea processes. This would not considerably change the optimization routines and their conclusions on $g(t)$, but it will affect the posterior uncertainties on $f(t)$. Inference is then still carried out on the RSL $g(t)+kj$ which is the main component of interest.

Following the reviewer's advice, we moved the fine-scale process variance (σ_u^2 , a.k.a. white noise) to the process level of the model, and the variance of the data, which the reviewer named σ_v^2 , is now the sole contributor to the sea-level observation errors, ϵ_i . All ϵ_i have been assigned a uniform standard deviation of 0.090 m, which is determined independently from an extensive survey of living microatolls (modern equivalents of our mid-Holocene coral microatolls) at our two sites. An assumption inherent in this is that the mid-Holocene microatolls had a similar spread of elevations as the modern microatolls, but this seems like a reasonable assumption, as the reef morphology shouldn't have changed significantly over that time, and it is unlikely that waves, currents, or sediment sources (which could move sand around and affect ponding) would have been significantly different 7000 years ago, either. We now devote a full paragraph to discussing this (under the sub-heading "Vertical uncertainties of microatoll data") in the Results section in the main text.

The process model:

- The temporal process (RSL) is modelled as a Gaussian process. Since the measurement time point is not exactly known, this is an error-in-variables problem. First, using ϵ for both the error in variables and measurement error is confusing, I would use a different symbol. Second, the authors state that “geochronological uncertainties were not incorporated into the hyperparameter optimization for computational efficiency.” I found this a bit surprising given that there was so much talk about the uncertainty of radio-carbon dating. Clearly there are uncertainties that can be defensibly ignored (e.g., intra-coral-slab measurements) and other that can not (intra-site slabs). The authors use 4 FCAs, but this is in fact a small number given all the ways in which the FCAs at the TKUB site can be shifted, and one may always argue that there are scenarios that weren’t explored. Crucially, considering only a few cases will not yield tenable uncertainty quantification assessments. I suggest the authors explore better ways of incorporating these uncertainties (these would go into ϵt) and do away with the use of subjectively chosen FCAs/binning methods. This would require some prior modelling assumption on the relationship between HLS and HLG states (e.g., using the sin curve above) and possibly MCMC, but the resulting inferences will be more defensible.

Again following the reviewer’s advice, we have changed our methods to do away with the FCAs. (Note that we have retained the concept of floating chronologies, which is useful and widely accepted, but we have removed the invented construction of “floating chronology arrangements” or “FCAs,” which were subjectively chosen arrangements of floating chronologies; in retrospect, we agree that the FCAs were problematic for all the reasons noted by the reviewer.) Instead of having discrete, subjectively chosen FCAs, we now optimize the shifts between floating chronologies as parameters in the model. Therefore, we incorporate geochronological uncertainties between the slabs (inter-slab uncertainties), while still ignoring the intra-slab measurement uncertainty. We introduced three new parameters: Δ_1 and Δ_2 (shifts within the TKUB site slabs) and Δ_0 (the shift between TKUB and TBAT). The first two parameters are the shifts between the three floating chronologies at TKUB and have a maximum sum of 21 yr and a minimum sum of 0 yr. This restricts the total temporal shift of the data, and allows us to optimize exact values for the shifts instead of restricting our model to arbitrary cases. The Δ_0 shift (between TKUB and TBAT) has a permitted range of ± 120 yr, consistent with the uncertainties in ΔR (see further discussion in the Methods). Note that we still treat TKUB-F04 and TKUB-F05 as a single floating chronology, and TKUB-F16 and TKUB-F19 as another single floating chronology; we argue we can do this because of the similar radiocarbon age estimates and the matching diedown chronologies within each pair of slabs. In addition, we have replaced the line of text quoted above with “Age uncertainties within individual floating chronologies are not incorporated into the model, as the law of superposition prohibits swapping the order of data derived from successive annual ϵ bands, effectively rendering the relative age uncertainty to be negligible.” Lastly, we now use only one ϵ , to represent the measurement uncertainty, and we use $w(t)$ to represent the white noise.

- The Taylor expansion in line 327 implies that $\text{var}(\epsilon_i^t)$ needs to be specified for each i , yet there is no talk of ϵ_i^t anywhere else in the text. Is it site-specific? How is it used? Was the variance of ϵ_i^t estimated or was it fixed to something reasonable? The authors did not use it for hyper-parameter estimation but one of the benefits of linearisation is that a linear-Gaussian model is recovered, and this should not upset the computational efficiency substantially.

This was a mistake. We are not using these errors, as we are not doing the geochronological uncertainty optimization nor the noisy-input GP routine. It seems we included this in the write-up mistakenly from a prior version of the model. We thank the reviewer for catching that!

- From the arguments presented in the text, I suspect the authors are not including any systematic component in ϵ_i^t so that they use FCAs instead. However, a more realistic model for ϵ_i^t would be $\epsilon_{i,j}^t = \mu_j + \zeta_i$ where μ_j captures the coral’s specific shift and ζ_i is just white noise. This, again, would allow the authors to do away with the FCAs and to obtain inferences on RSL that capture all of the uncertainty (plus, one would get objective information on the specific shift μ_j). Note that μ_j is in units of years and is different

from k_j , which is in mm, is site-specific, and which I imagine should be well-constrained. I am not convinced linearisation is needed for computational efficiency in this problem, even if a systematic component is considered: I suspect McHutchon and Rasmussen (2011) developed this model for when one has multiple input dimensions, of which the authors only have one. I briefly show an example where I try to estimate the systematic component from the data at the end of this section; adding noise with known variance to this systematic component would be straightforward.

Although we did not implement a fully-Bayesian model, we did optimize these shift, and we have moved the white noise to the process level, as suggested, to the model.

- The notation needs some improving. In line 334, replace $g(t_i)$ with $g(t)$. Also in line 327 f should be site-specific (i.e., use f_j), and $s(t)$ on page S.21 is also defined to be site-specific.

We have revised the notation to make it more clear.

The cross-validation:

- The cross-validation exercise is an important one. However I have my concerns with the use of all the limiting data with the proposed model; please see these above. The authors have constructed a score (based on the cross-validated likelihood, or log predictive score) using all data points. Using all the data points is non-standard, since one is including training data when assessing model fit. I suspect that using all the data in the scoring will not penalize for over-fitting as much as one would wish. Why not carry out a typical cross-validation exercise and just validate using the left-out data? Also, it would be good to show other diagnostics of the cross-validated data such as the mean absolute prediction error or some hybrid score such as the continuous ranked probability score. If this cross-validation procedure was taken from a book or paper please provide a citation (general note: Please provide more citations for statistical methods employed).

With our new model, we have implemented a “typical” Leave-One-Out Cross Validation (LOOCV) of the model, as the reviewer suggested. We have also tried to add in more citations where appropriate.

- I found the notation in this section a bit difficult to follow. Sometimes P is used to denote a probability, and sometimes a probability density function. Also the reader is left to guess what many things are. For example, is f_L the trained process evaluated at the limiting data or the process trained with the limiting data? L is a set since the authors use $i \in L$; what is this set? I suggest this section is revised to make the notation clearer.

We have attempted to make this clearer. We apologize for all the ambiguities the first time around.

- I am not sure what the final paragraph in 6.2 is suggesting. In particular, I don't see how one can validate against the true signal $f(t)$ if this is not known. Since one only has observations y , then the error term needs to be included in the likelihood when validating. White noise contains no information, and will not contribute to predictive performance in the mean. I suspect the authors have made this choice because they are using all the data (including the training data) in the cross-validation. This would not have been an issue if only the validation data was used, as is typically the norm when assessing model fit.

By using a different model and a standard cross validation, we have eliminated this problem, and the entirety of Section 6 of the supplement in the previous version has been replaced by a single paragraph describing the cross-validation exercise, now Section 3.

- In Table S5 it is important that the math symbols are used as the labels are subject to interpretation (what is σ with no subscript? Is it a confidence interval? Does ‘Time scale’ refer to τ ? If so why are the units mm and not yr?). Note that if the HLS and HLG relationship is modelled and the shifts are estimated (which I believe is possible, see below) then this table would collapse to one row.

σ with no subscript was supposed to be a confidence interval, and the units for the timescale was a mistake—we apologize for the confusion. That said, Table S5 in the previous version has been replaced by Table S3, and as the reviewer anticipated, it has collapsed to a single row. (We show two rows in the present Table S3, because we also provide information for the “alternative” model, described above, in which we use nearly all of the limiting data.)

Use of MCMC in estimating intra-site carbon dating errors

I believe the authors can do away with linearity assumptions on the error and should attempt a full analysis without FCAs. I did a quick test to check whether this is indeed possible. Let $x(t)$ be a Gaussian process with a Matérn correlation function with smoothness parameter $3/2$ as defined by the authors. Then assume that $y_1(t) = x(t) + \varepsilon_1$ and $y_2(t) = x(t + \Delta) + \varepsilon_2$. In their case $y_1(t)$ would be the measurements at the TBAT site and y_2 would be data from one of the slabs at TKUB. For this simple example I assumed that the process parameters of $x(t)$ are known, that $\Delta = 50$, and that the variance of ε_1 and ε_2 was equal to 1. I probably have used more observations than the authors have available but also threw in substantial measurement error.

In order to infer Δ I used MCMC, where I assumed that Δ was a priori normally distributed centred at zero. I obtained 2000 samples in a few seconds on my machine, and I don't think the authors have so much data that this will become computationally unfeasible. In Fig. 1 I show the true process $x(t)$ (unobserved), y_1 and y_2 . In Fig. 2 I show the posterior distribution on Δ after removing the first 400 samples for burn-in. This indicates that it is possible to propagate uncertainties in the carbon dating all the way through. I'm not suggesting this will be as easy to implement in the author's application, but not properly accounting for these uncertainties may influence the inferences, and this is of concern in error-in-variables problems.

We thank the reviewer for this suggestion, and we appreciate the reviewer's efforts to test whether this would have been computationally feasible. In the end, we accomplished the full analysis suggested by the reviewer by incorporating and optimizing an additional delta (shift parameter) for the shift between TBAT and TKUB.

Reviewer #3 (Remarks to the Author):

Review of Meltzner et al by Alex Thomas

The authors present a new and substantial dataset of proxy sea level from southeast Asia covering a brief time interval from 6800 to 6500 years BP. These data are highly relevant as they provide insight into the potential for multi-decadal sea level variability during the Holocene. Such variability is currently highly contested and would therefore be of interest to a wide range of researchers interested in both paleo and modern sea level variability.

The data comprise the reconstruction of past sea level marker points from coral microatolls and combined radiocarbon and layer counting chronologies. The data are from two independent sites and qualitatively agrees with existing data from the region adding to the confidence in the results. This is an established method for reconstructing sea level but this study provides new data from a previously understudied region.

I feel that the results are robust but there are some points of clarification required regarding the maximum rates of change and some places where the manuscript could be improved prior to publication.

We thank Dr. Thomas for recognizing the importance of our study and for all of his suggestions, below.

The authors have done a thorough job in constraining the uncertainties in the elevations of the RSL markers and in constructing the age models. The age models within each coral microatoll are highly precise due to the constraint of layer counting but the absolute ages and ages between sites have relatively large uncertainties due to the inherent uncertainty of marine radiocarbon age determinations. The authors are careful in their interpretation not to interpret the relative ages of the sea level changes at different sites and being at the same time, but the limitations of the chronology should be better accounted for in the main text. The uncertainty in the timings of the changes in RSL between the different samples is consistent with the sea level changes happening at the same time but it could be that there is a real difference in timing. This should be more clearly acknowledged and the implications for the potential mechanisms of regional sea level variability discussed.

We thank the reviewer for this suggestion. Following this advice, we moved the discussion of chronological constraints and uncertainties from the Methods to the Results section of the main text. For precisely this reason, we present the coral data twice in the main text: First, on Figs. 2–3, we present the data with their original radiocarbon-based dates, and we emphasize all the dating uncertainties on the figures and in their respective figure captions. Second, on Fig 4, we present the age-modeled time series, in which the dates of the various floating chronologies at the TKUB site have been optimized by the empirical hierarchical model. This provides more transparency in our methodology, and it reminds readers that a real difference in the dates is possible (albeit unlikely). In addition, our model now optimizes the timing of the various floating chronologies, making it more objective than before, and we can now assess the quality of the fit in a more statistically rigorous manner.

The rates of change of sea level seem to be inconsistent between the figure and the text. The text claims rates of rise and fall for 18.1 and -23.3 mm/yr (also units are inconsistent some figures used m/kyr) [lines 126-127], but the rates illustrated in Figure 3 (and in the SI) show much lower maximum rates of around 7 and -20. I can't figure out where this discrepancy arises. But this must be reconciled before publication. Additionally there when combining datasets to produce a single RSL curve with a probabilistic model then the more data sources that are included then there will be a smoothing artefact if there is any uncertainty in the age models between the datasets. The rates calculated are therefore likely an underestimate. It would therefore be nice to see the rates calculated for the individual sites as well as the combined.

We apologize for the confusion. The apparent inconsistency arose because we were using the four different FCAs, and we determined different rates for each FCA; then, when stating the rates in the prose, we tried to simplify and summarize the results. The FCAs were highly problematic for other reasons (see our response to Reviewer 2), so we have eliminated those, and both our calculation and our discussion of rates is now more straightforward. Supp. Table S4a (with rates averaged over 20-yr running windows) should now be consistent with Supp. Figs. S11e and S12c, and with the text. (Note that we prefer the model in Supp. Fig. S11 and discount the rates from Supp. Fig. S12 as not credible, as discussed in the Methods; hence, the rates mentioned in the abstract reflect only those from Supp Fig. S11.) Also, for units in the main text, we now stick to meters for length (except when mentioning the 80 km or 2600 km distances between the sites, for which we use kilometers) and mm/yr for rates. We feel these choices allow us to consistently use numbers that do not have too many decimal places. We also thank the reviewer for pointing out the issue of rates likely being an underestimate in cases such as this. Unfortunately, it is not possible to calculate separate rates for individual sites without changing the model itself; furthermore, the relative timing of the floating chronologies at TKUB is constrained both by radiocarbon ages and by data from TBAT, so the two datasets are interdependent, at least when it comes to fine-tuning the age models. In the end, the biggest variable in determining the peak rate of RSL fall is the relative timing of the two older floating chronologies at TKUB. At the end of the “Shared RSL curve for TKUB and TBAT” section in the Results, we added the following sentence: “If the 21-yr shift between the floating chronologies at TKUB were reduced as contemplated in the previous paragraph, the peak rate of RSL fall (~6770 cal yr BP) would be even faster.” We hope this adequately addresses the reviewer’s concern.

Perhaps the most disappointing part of the discussion for me is the interpretation of the changes in regional sea level. There seems to be little critical analysis of the potential causes of the variability. The amplitude of 60-70 cm seems large compared to the variability of ENSO or the monsoon. How much of the observations could be explained by these mechanisms? Additionally does the timescale of the changes to fit with known ENSO variability there are hints of similar timescales in the coral data of Cobb et al 2003. Dose the spatial patterns of the amplitudes of the variability match for the three records we have versus that expected for ENSO driven or global ocean volume driven changes?

We have struggled with this part of the manuscript, because of a lack of information or models that would allow us to quantify expected changes in sea level in our study areas due to persistent climate oscillations. We thank the reviewer for pointing out the Cobb et al (2003) study, which is an important addition to our Discussion, and we now relate it to regional sea-level variability over decadal timescales, related to ENSO and the PDO, as observed by satellites, and as projected for the coming century. The reviewer's last question is particularly difficult to answer: as shown in Supp. Fig. S15, the amplitude hyperparameter for the Leizhou Peninsula dataset is poorly determined, leaving some (nontrivial) uncertainty in the amplitude of the sea-level fluctuations at the Leizhou Peninsula site. Given that uncertainty and the uncertainties in what would be "expected" in various scenarios, more than one answer would be possible.

Figure 2 the grey bands showing the RSL histories at each site are clear but it would be good to include in that the key data which is used to constrain each curve (at least for the new sites. This is shown in the SI but it would make it much clearer for non expert readers hoe the curves were constructed and their relative merits. The key points are the highest level of survivals which could be made more obvious than they are in the SI figures.

The data are now plotted with the curves in Fig. 4. Using a combination of the larger circles for HLS points (as in Figs. 2–3) and the different style of error bars (from the SI figs), we feel the HLS data (index points) are now more obvious.

Line 105 which way would the tidal biases bias the relative amplitudes?

Immediately following the sentence cited by the reviewer, we have added the following text to the manuscript: "The tidal range at both sites is modeled to have been slightly higher at 6–7 kyr BP, implying that mean sea level was up to ~0.1 m higher than shown on Figure 4; this effect would have been more pronounced at TBAT than at TKUB."

Line 107 "Preclude" means to prevent, 'exclude' would be more appropriate.

We have changed this as suggested.

The title is not strictly correct. For this variability to be "oscillations" requires a defined frequency and amplitude. Neither of these is strictly confirmed. "Vacillations" would be more correct but somewhat esoteric. "Rapid sea level variability on societally relevant timescales"?

We prefer "fluctuations," which does not require a defined frequency or amplitude. We hope this is acceptable to the reviewer and the Editors.

REVIEWERS' COMMENTS:

Reviewer #1 (Remarks to the Author):

I am happy that the authors have addressed all concerns and recommend publication without further revision.

Reviewer #2 (Remarks to the Author):

The authors have taken into consideration my previous comments and the new manuscript rectifies the concerns I had earlier.

Reviewer #3 (Remarks to the Author):

This is clearly a much more clearly presented manuscript and there has been a much more rigorous interpretation of the data than in the previous iteration that I saw.

My requests for a wider discussion of the regional sea level changes that might have led to these results and the implications for regional climate variability have not been totally addressed in this new version but this would perhaps be beyond the scope of this data intensive paper and would be more suited to a longer discussion piece.

I recommend this paper for publication